# ALL IN RLVR ON NON-VERIFIABLE DOMAINS

## ABSTRACT

Reinforcement Learning with Verifiable Rewards (RLVR) has demonstrated great success on verifiable domains such as math and coding abilities for large language models (LLMs). However, open domains, with their subjectivity and lack of ground truth—have long been considered fundamentally challenging for RLVR, limiting its application. In this work, we challenge this view and pioneer a novel methodology to extend RLVR into open domains. We first reveal that a *partial* and imperfect sample-level reward is sufficient for RL under certain conditions. On the top of this discovery, we introduce sample-specific *Judge Code* as programmatic rubrics, which replaces the traditional reward model (RM) to evaluate LLM responses. Specifically, our methodology is centered on a Judge Code Generator (JCG), which programmatically translates evaluation rubrics into executable Judge Code for each sample. Judge Code serves as a partial and computationally efficient instantiation of the evaluation rubric. The system supports two operational modes: in Online mode (On-JCG), it dynamically generates (`Query`, `Judge Code`) pairs on-the-fly to create a reusable dataset for subsequent RL training; in Offline mode (Off-JCG), it directly leverages this pre-generated dataset to enable highly efficient, RM-free training. Through experiments, we demonstrate the promising potential of applying RLVR methods to open domains. Moreover, we particularly emphasize one of the key benefits brought by efficiency: compared with RM-based methods, specifically the generative reward model (GenRM), Off-JCG achieves more than 2x speedup in wall-time when reaching competitive performance. This work highlights a promising direction of reshaping the understanding of RLVR and open-domain research.

## 1 INTRODUCTION

The integration of Reinforcement Learning (RL) with Large Language Models (LLMs) has emerged as one of the most important techniques in post-training and alignment (Ouyang et al., 2022; Bai et al., 2022). Since the emergence of DeepSeek-R1 (Guo et al., 2025a), Reinforcement Learning with Verifiable Rewards (RLVR) has achieved remarkable success, demonstrating substantial improvements in reasoning capabilities across verifiable domains like math, coding, and sandbox agents (Yu et al., 2025a;c; Jin et al., 2025). Moreover, RLVR is believed to have powerful scalability and generalization capabilities (Guo et al., 2025a; Chu et al., 2025; Deng et al., 2025).

However, despite the advances of Reinforcement Learning with Verifiable Rewards (RLVR) in verifiable domains, broader open-domain applications have predominantly relied on Reward Models (RMs). The core challenge lies in evaluation: due to the inherent subjectivity and lack of a single ground-truth answer in open-domain tasks, designing objective reward signals is exceptionally difficult (Zhang et al., 2025a). Therefore, in the context of this work, we refer to this class of difficult-to-verify open domains as "Non-verifiable domains", contrasting them with traditional verifiable domains — and we boldly challenge the aforementioned

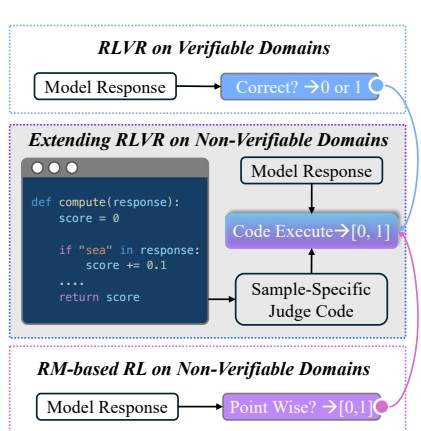

Figure 1: Our method extends RLVR on non-verifiable domains through sample-specific Judge Code .

overreliance on RMs by pioneering a simple methodology to fully extend RLVR into these non-verifiable open domains. Rather than some related works which might incorporate rule-based rewards as a minor part of the evaluation criteria for open domain tasks, such as answer format, length requirements, and keyword constraints (Guo et al., 2025a; Peng et al., 2025), we "all in" RLVR on non-verifiable domains by taking Judge Code to evaluate open tasks. This unified framework would help us to adapt those proven RLVR techniques from verifiable domains to non-verifiable domains.

The key insight enabling RLVR application to open domains lies in our discovery. We discover that a *sample-level* partial and imperfect reward signal in each step is sufficient for effective RL training, achieving performance ceilings nearly identical to those obtained with more nuanced reward signals. This holds provided that the multiple partial rewards collectively ensure complete *dataset-level* coverage and diversity. This finding has been validated through our pilot studies on multi-instruction following tasks, where we demonstrate that randomly picking only one rule per training instance and providing corresponding rewards enables the model to learn all rules through extensive training iterations. We extend this discovery to open domains by employing sample-specific Judge Code as a partial, imperfect reward signal.

The Judge Code is generated by what we term a Judge Code Generator (JCG), an LLM Coder. For each sample, JCG generates sample-specific Judge Code as a programmatic rubric, which takes the actor model's response and outputs the score. While each individual Judge Code only captures a small and partial subset of the desired behaviors or quality criteria for a given sample, the collective effect across the entire dataset is expected to provide sufficient coverage and diversity to guide the model toward improved performance. Our methodology supports two distinct training modes: On-JCG, which integrates the JCG into the training pipeline to generate Judge Code in real-time based on prompts and actor model responses, and the generated (`Query`, `Judge Code`) can be stored; and Off-JCG, where it directly leverages the pre-generated dataset for subsequent RL training, achieving truly RM-free RL.

Unlike some recent works about unsupervised RL methods that rely on internal confidence measures (Zhou et al., 2025; Yu et al., 2025b; Zuo et al., 2025), we do not need a reference for open-ended tasks. Instead, we still leverage an external but weak reward based on Judge Code. We acknowledge that this approach is widely adopted in RL for code generation, but our key contribution is extending and adapting it to non-verifiable open domains.

Our extensive experiments demonstrate that this streamlined approach not only enhances model capabilities in open domains but also provides a unified perspective for RLVR across all domains. Notably, our Off-JCG approach eliminates the dependency on RM entirely while achieving performance nearly equivalent to traditional RM-based methods and delivering over 2x faster training speed. This represents both a conceptual breakthrough that challenges conventional paradigms and a solution with significant practical value.

The success of our method is not counterintuitive; rather, it is grounded in the fundamental principles of stochastic optimization. Our theoretical analysis (Section 5.2) shows that the partial reward signal, while noisy for any single sample, functions as a perfect unbiased estimator of the complete reward. This ensures that the policy gradient, in expectation, points in the correct optimization direction. Consequently, the learning process effectively averages out the sample-level noise, guiding the policy towards the same optimum as would be found using the full, often intractable, reward signal. We contend that applying RLVR to non-verifiable domains represents an emerging and promising research direction.

In summary, the contributions of this paper are threefold:

1. We challenge the conventional belief that RLVR is incompatible with open-domain tasks and propose a comprehensive methodology to successfully apply RLVR across diverse domains, achieving performance comparable to traditional reward model approaches.

2. We demonstrate that sample-level partial and imperfect reward signals are sufficient for effective RL training, provided that there is adequate dataset-level coverage and diversity, thereby eliminating the need for nuanced reward design in open domains.

3. We introduce the Judge Code Generator (JCG) with On-JCG and Off-JCG training modes, enabling truly RM-free RL while maintaining competitive performance across diverse open-domain applications.

> Can you provide a sample journal entry focused on time management techniques? Explain your thought process in creating the example. ①Ensure that in your reply, the character 'k' is included less than 8 instances. ②Ensure that every part of your response is written exclusively in English. Do not include any text in other languages. ③Respond with less than 12 statements. ④Ensure the response includes 3 placeholders enclosed in square brackets, for example: [address]. ⑤In your response, ensure you include precisely 3 bulleted items. Format each point using markdown bullets, like this: * Example point here. ⑥Give two different responses. Responses and only responses should be separated by 6 asterisk symbols: ****** ⑦Ensure that terms written in uppercase letters occur at least 6 instances in your reply. ⑧Wrap your entire response with double quotation marks. ⑨In your entire response, refrain from the use of any commas.

Figure 2: The high-complexity prompt used in Phase 1 of our pilot study. It includes nine distinct, verifiable constraints designed to test the model's ability to handle multifaceted instructions.

## 2 PILOT STUDY: VALIDATING THE EFFICACY OF PARTIAL REWARDS

To establish the foundational principle of our work—that effective training does not require a complete, holistic reward signal—we conducted a two-phase pilot study. The goal was to investigate the relationship between instruction complexity and reward signal density, and their impact on the final performance of Qwen2.5-7B (Qwen et al., 2025). All experiments in this study were conducted using a Chain-of-Thought (CoT) template (see Appendix C) to elicit the model's reasoning process.

**Phase 1: Training on a Single, Complex Task.** We trained a model on a single, complex 9-constraint prompt (Figure 2) to assess learning under three distinct settings: 1) Full Reward (9/9), a baseline where the model received the full prompt and was rewarded on all nine constraints; 2) Partial Reward (1/9), where the model received the full prompt but was rewarded for only one randomly selected constraint; and 3) Simplified Task (1/1), where the model received a decomposed prompt (split from the original complex prompt into nine simpler, single-constraint prompts) and was trained on one of these single-constraint tasks at a

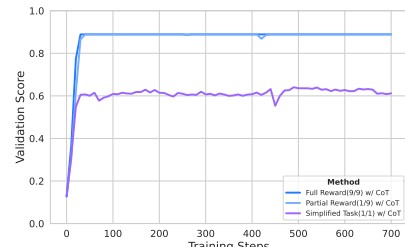

Figure 3: Phase 1 Results.

time. As shown in Figure 3, the Partial Reward setting's performance nearly matched the Full Reward baseline. In contrast, the Simplified Task was significantly inferior, suggesting that exposure to the full task context is crucial for learning to satisfy all constraints simultaneously.

**Phase 2: Training on Augmented Tasks.** To extend the findings from Phase 1 and explore the stability of the observed effects across expanded task scopes, we constructed Aug-IFEval, a series of new datasets. Following the methodology of IFEval (Zhou et al., 2023), we generated prompts via a systematic process: 1) categorizing a large pool of instructions to avoid logical conflicts, 2) using an LLM to combine and rephrase them into a diverse set of 7-constraint prompts, and 3) creating four distinct training datasets from this master set for different reward densities. This process enabled a large-scale comparison of a Full Reward (7/7) regime against three

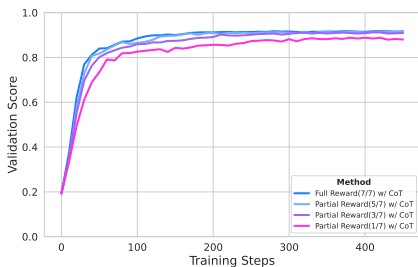

Figure 4: Phase 2 Results.

Partial Reward settings (5/7, 3/7, and 1/7), where a random subset of constraints were evaluated. The results (Figure 4) robustly replicated our initial findings: all Partial Reward settings, even the sparsest (1/7), achieved performance comparable to the Full Reward baseline with no significant degradation.

**Conclusion of Pilot Study.** This two-phase pilot study provides compelling evidence for our core thesis. A *sample-level* partial and imperfect reward signal in each step is sufficient for effective RL training. This lays the groundwork for our main experiments, motivating the development of a scalable method for synthesizing such effective partial rewards for non-verifiable domains.

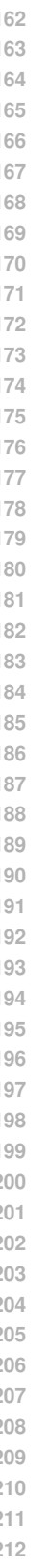

Judge Code Generator-Online Training

Judge Code Generator-Offline Training

Figure 5: **Overview of the Judge Code-guided Reinforcement Learning (JC-RL) framework.** The On-JCG phase (top) involves three steps: the actor model generates responses, the Judge Code Generator creates a query-specific Judge Code $C_{q_t}$, which is executed to compute rewards. This phase produces the offline dataset $\mathcal{D}_{\text{offline}}$, consisting of (Query, Judge Code) pairs. The Off-JCG phase (bottom) leverages this pre-generated dataset for efficient, low-cost RL training by skipping the expensive code generation step.

## 3 METHODOLOGY

Our pilot study (Section 2) established a key principle: a *sample-level partial reward* is sufficient to guide reinforcement learning. While this finding obviates the need for a perfect, holistic reward function, it introduces a new challenge: *how can we systematically generate partial rewards for complex, open-ended tasks where rubrics are ambiguous or infinite?* To address this challenge, we propose the Judge Code-guided Reinforcement Learning (JC-RL) framework. This process can be viewed as a form of knowledge distillation, where a powerful Judge Code Generator's broad understanding of quality is distilled into a set of weak, programmatic rubrics. These rubrics are instantiated as executable **Judge Code**, transforming the abstract problem of reward design into a concrete engineering task (its characteristics are analyzed in Section 5). As illustrated in Figure 5, JC-RL is a dual-phase system which includes both an online Judge Code Generator-Online (**On-JCG**) phase and a highly efficient Generator-Offline (**Off-JCG**) phase for RL training.

### 3.1 ON-JCG TRAINING

The primary objective of the online phase is to generate a query-specific, executable Judge Code as a reward function for each query, thereby constructing a high-quality, reusable dataset. This process, for each query $q_t$, is divided into three stages:

1. **Rollout Phase:** The current actor model $\pi_\theta$ generates a set of $n$ diverse responses $\mathbf{Y}_t$ for a given query $q_t$:
$$\mathbf{Y}_t = \{y_1, y_2, \ldots, y_n\}, \quad \text{where } y_i \sim \pi_\theta(\cdot|q_t) \tag{1}$$

2. **Judge Code Generation Phase:** Instead of direct scoring, we employ a Judge Code Generator ($\mathcal{G}$), a powerful model that generates programmatic rubrics in code form. It takes the query $q_t$, the set of generated responses $\mathbf{Y}_t$, and an induction-prompt template $\mathcal{T}_{\text{induction}}$ (see Figure 13) as input to generate a unique, executable Judge Code ($C_{q_t}$):
$$C_{q_t} = \mathcal{G}(q_t, \mathbf{Y}_t, \mathcal{T}_{\text{induction}}) \tag{2}$$

This code serves as the programmatic realization of the rubric, encoding the core evaluation logic for query $q_t$. A crucial output of this phase is the collection of (Query, Judge Code) pairs, which constitute our offline dataset $\mathcal{D}_{\text{offline}}$:
$$\mathcal{D}_{\text{offline}} = \{(q^{(j)}, C_q^{(j)})\}_{j=1}^M \tag{3}$$

3. **Reward Computation Phase:** The generated Judge Code $C_{q_t}$ is executed within a secure sandbox to score responses $y_i \in \mathbf{Y}_t$. This produces a reward vector $\mathbf{R}_t$:

$$R_{\text{JC}}(y_i|q_t) = C_{q_t}(y_i, q_t) \tag{4}$$

$$\mathbf{R}_t = [R_{\text{JC}}(y_1|q_t), \ldots, R_{\text{JC}}(y_n|q_t)] \tag{5}$$

## 3.2 OFF-JCG TRAINING

The offline phase leverages the pre-generated dataset $\mathcal{D}_{\text{offline}}$ to fine-tune any actor model $\pi_\theta$ with high efficiency. A key advantage of this phase is that the Judge Code has already been generated during the online phase—thus, the Judge Code Generation step is skipped entirely, eliminating the computational overhead of on-the-fly code generation. The workflow follows a two-stage process (adapted from the online phase's three-stage framework) with the following steps:

1. **Rollout Phase:** Identical to the online phase, where the actor model $\pi_\theta$ generates responses $\mathbf{Y}_t$ for a query $q_t$ sampled from $\mathcal{D}_{\text{offline}}$.
2. **Reward Computation Phase:** Identical to the online phase, where the paired $C_{q_t}$ is executed to compute rewards $\mathbf{R}_t$ and update the model.

This offline approach significantly reduces computational costs by bypassing the most expensive Judge Code generation step while maintaining almost the same reward quality.

## 4 EXPERIMENTS

### 4.1 SETUP

**Models and Environment.** For our primary actor models, we employ Qwen2.5-7B (Qwen et al., 2025) and Llama-3.1-8B (Dubey et al., 2024). The Judge Code Generator ($\mathcal{G}$) is instantiated using the DeepSeek-V3 (DeepSeek-AI, 2024). To ensure consistently high inference performance, this model was self-deployed on servers equipped with NVIDIA Hopper series GPUs. We selected the `VeRL` (Sheng et al., 2024) to orchestrate our experiments due to its robust support for distributed training on our NVIDIA Hopper series GPU environment and its flexibility in integrating custom, non-differentiable reward sources like our programmatic generator. We utilize the DAPO algorithm (Yu et al., 2025a), a variant of GRPO Shao et al. (2024), for optimization. Our hyperparameters follow standard practices and include sampling temperature of 0.7, with KL-divergence regularization disabled. We use a training batch size of 256, and during the rollout phase, $n = 8$ responses are generated per prompt. The actor model is trained with a learning rate of $1 \times 10^{-6}$. We set prompt length as 2048 tokens and response length as 4096 tokens, which exceeds the requirements of our dataset. Our training setup is a simplified configuration inspired by (Xie et al., 2025) to improve reasoning capabilities, and a standardized Chain-of-Thought (CoT) template (Appendix C) is consistently applied across all experiments.

**Datasets.** Our dataset selection focuses on performance in complex and subjective scenarios:

- **Non-Verifiable Domain Tasks:** We utilize `No Robots` (Rajani et al., 2023), `WritingPrompts` (Fan et al., 2018) (creative writing), and `ShareGPT`[1] & `WildChat` (Zhao et al., 2024) (real-world dialogues including diverse conversations).
- **Verifiable Domain Tasks:** For an objective performance reference, we use the `Aug-IFEval` (instruction following) dataset constructed in Section 2.

**Evaluation Benchmarks and Metrics.** To comprehensively assess model capabilities across diverse tasks, we employ several established benchmarks, reporting the most appropriate metric.

- For models trained on the `WritingPrompts` dataset, we evaluate creative writing prowess using the Creative Writing Benchmark v3, reporting the Rubric Score (0-100) as the primary metric for absolute quality.

---

[1]`https://sharegpt.com/`, dataset version that was used: `https://huggingface.co/datasets/shibing624/sharegpt_gpt4`.

Table 1: The results of our primary experiments. The values in parentheses with colored backgrounds represent the absolute uplift over the base model. The color intensity corresponds to the magnitude of the improvement, with deeper shades indicating a larger gain.

| Training Set | Method | Qwen2.5-7B | | Llama-3.1-8B | |
|---|---|---|---|---|---|
| | | *Creative Writing v3* | | *Creative Writing v3* | |
| WritingPrompts | Base Model | 62.17 | | 42.12 | |
| | + GenRM | **72.72** (+10.55) | | **68.47** (+26.35) | |
| | + On-JCG (Ours) | 71.11 (+8.94) | | 63.22 (+21.10) | |
| | + Off-JCG (Ours) | 70.05 (+7.88) | | 55.89 (+13.77) | |
| | | *AlpacaEval 2.0* | *MT-Bench* | *AlpacaEval 2.0* | *MT-Bench* |
| No Robots | Base Model | 7.51 | 2.55 | 4.44 | 2.65 |
| | + GenRM | **36.79** (+29.28) | 5.40 (+2.85) | 10.10 (+5.66) | **6.62** (+3.97) |
| | + On-JCG (Ours) | 31.05 (+23.54) | **6.13** (+3.58) | **13.35** (+8.91) | 6.33 (+3.68) |
| | + Off-JCG (Ours) | 29.67 (+22.16) | 5.96 (+3.41) | 9.04 (+4.60) | 5.95 (+3.30) |
| ShareGPT | Base Model | 7.51 | 2.55 | 4.44 | 2.65 |
| | + GenRM | **27.29** (+19.78) | **6.05** (+3.50) | 9.75 (+5.31) | **6.37** (+3.72) |
| | + On-JCG (Ours) | 22.25 (+14.74) | 5.87 (+3.32) | 9.96 (+5.52) | 6.22 (+3.57) |
| | + Off-JCG (Ours) | 22.18 (+14.67) | 6.01 (+3.46) | 8.79 (+4.35) | 5.89 (+3.24) |
| WildChat | Base Model | 7.51 | 2.55 | 4.44 | 2.65 |
| | + GenRM | **28.11** (+20.60) | **4.45** (+1.90) | **6.77** (+2.33) | **6.04** (+3.39) |
| | + On-JCG (Ours) | 16.56 (+9.05) | 5.87 (+3.32) | 6.01 (+1.57) | 5.69 (+3.04) |
| | + Off-JCG (Ours) | 20.97 (+13.46) | 5.81 (+3.26) | 5.13 (+0.69) | 5.35 (+2.70) |
| | | *IFEval* | | *IFEval* | |
| Aug-IFEval | Base Model | 24.40 | | 9.24 | |
| | + Rule-based | **80.41** (+56.01) | | **81.70** (+72.46) | |
| | + On-JCG (Ours) | 70.61 (+46.21) | | 72.64 (+63.40) | |
| | + Off-JCG (Ours) | 69.69 (+45.29) | | 63.96 (+54.72) | |

- For general-purpose conversational abilities, models trained on `No Robots`, `ShareGPT`, and `WildChat` are evaluated on two widely-recognized benchmarks: AlpacaEval 2.0 and MT-Bench. For AlpacaEval 2.0, we report the length-controlled win rate, a metric known for its high correlation with human preferences and robustness against length bias (DeepSeek-AI, 2024). For MT-Bench, we calculate the average score, computed from the single-answer grades obtained from the two turns of each dialogue.

- For the verifiable instruction-following task, models trained on `Aug-IFEval` are evaluated on the held-out test set of IFEval (Zhou et al., 2023), reporting the strict accuracy.

**Reward Computation Mechanism.** Our reward computation follows a gated mechanism, prioritizing structural format over content quality. Formally, the reward function is defined as:

$$R(y_i|q) = \begin{cases} 1.0, & \text{if } \texttt{IsFormatValid}(y_i) = \text{False} \\ C_q(\texttt{ExtractAnswer}(y_i), q), & \text{otherwise} \end{cases} \quad (6)$$

Each response $y_i$ is first validated for its format using the `IsFormatValid()` function, which verifies the presence of required tags (e.g., `<think>` and `<answer>`). Responses failing this check are assigned a floor score of 1.0, effectively penalizing format errors. If the format is valid, the answer extracted by `ExtractAnswer()` is then evaluated by the Judge Code function $C_q$. This Judge Code is dynamically synthesized for each query $q$ via a universal, reasoning-focused induction prompt (see Figure 13). It operates by adding points for positive criteria (e.g., fulfilling requests, creativity) and deducting points for negative criteria (e.g., verbosity, factual errors). Its execution yields a final content score normalized to the range of $[1.0, 10.0]$.

**Baselines.** We compare our proposed On-JCG and Off-JCG modes against: **Base Model**, **GenRM** (DeepSeek-V3 as the generative RM), and **Rule-based** (for the verifiable domain).

## 4.2 EFFECTIVENESS AND EFFICIENCY

As shown in Table 1, our proposed On-JCG and Off-JCG methods achieve significant performance gains with such a weak and partial reward.

**Effectiveness.** In open-domain tasks, both On-JCG and Off-JCG substantially outperform the Base Model. Notably, the performance of Off-JCG is competitive with the costly GenRM baseline, confirming Judge Code as a highly reliable reward signal. In the verifiable domain, our methods also provide considerable improvements, albeit not reaching the ceiling set by the ideal Ground-Truth Reward baseline.

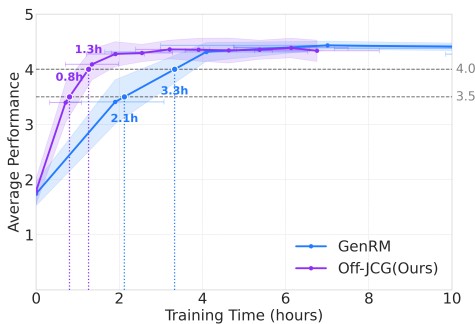

Figure 6: Off-JCG achieves **2x faster** speedup than Generative RM.

**Efficiency.** The Off-JCG mode exhibits a compelling advantage in training efficiency. As illustrated in Figure 6, our offline method achieves performance levels close to GenRM but more than 2x faster, highlighting its value for rapid and economical training in most non-verifiable domain tasks (see Appendix F for complete experimental details).

## 4.3 GENERALIZATION OF OFFLINE DATASETS

A core claim of our work is that the (Query, Judge Code) dataset ($\mathcal{D}_{\text{offline}}$), our reusable offline asset, is not merely a data collection but a portable, model-agnostic, and scale-agnostic knowledge base. To substantiate this, we conducted two sets of generalization experiments (Figure 7).

**Cross-Model Generalization.** We first tested whether the reward logic is tied to the source model's architecture. We successfully fine-tuned a Llama-3.1-8B model using the offline dataset ($\mathcal{D}_{\text{offline}}$) from Qwen2.5-7B, and vice-versa. As shown in Figure 7a and 7b, both models achieved significant performance gains, proving that the encoded reward logic is universal and effectively transferable across different model families.

**Cross-Scale Generalization.** Next, we investigated whether assets generated with smaller models could align larger ones. We applied the $\mathcal{D}_{\text{offline}}$ from 7B models to fine-tune larger 14B and 32B models. The results (Figure 7c and 7d) are compelling: the more capable models leveraged the same reward signals to achieve even higher performance ceilings. This confirms the excellent scalability of our offline assets.

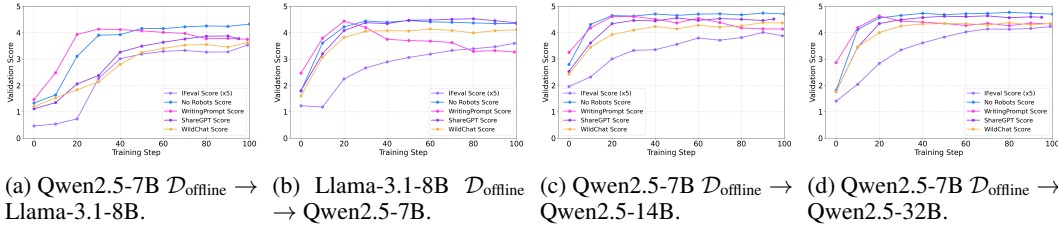

(a) Qwen2.5-7B $\mathcal{D}_{\text{offline}} \rightarrow$ Llama-3.1-8B.

(b) Llama-3.1-8B $\mathcal{D}_{\text{offline}} \rightarrow$ Qwen2.5-7B.

(c) Qwen2.5-7B $\mathcal{D}_{\text{offline}} \rightarrow$ Qwen2.5-14B.

(d) Qwen2.5-7B $\mathcal{D}_{\text{offline}} \rightarrow$ Qwen2.5-32B.

Figure 7: Generalization across models and scales. (a, b) demonstrate **cross-model generalization**. (c, d) demonstrate **cross-scale generalization**.

## 4.4 CODE SCALING: THE IMPACT OF AUXILIARY RUBRICS ON TRAINING DYNAMICS

Beyond validating necessary components, we explored the possibility of enhancing our framework through "code scaling"—the introduction of an additional, auxiliary Judge Code generated via another induction prompt (as shown in Figure 11) into the reward computation process to enable a form of "discussion" or adversarial checking.

Table 2: Code Scaling Study: Impact of the Auxiliary Judge Code on **Off-JCG** Performance.

| Training Set | Method | Qwen2.5-7B | | Llama-3.1-8B | |
|---|---|---|---|---|---|
| | | *Creative Writing v3* | | *Creative Writing v3* | |
| WritingPrompts | Off-JCG | 70.05 | | 55.89 | |
| | + w/ Aux Code | **71.21** | | **56.88** | |
| | | *AlpacaEval 2.0* | *MT-Bench* | *AlpacaEval 2.0* | *MT-Bench* |
| No Robots | Off-JCG | **29.67** | **5.96** | **9.04** | 5.95 |
| | + w/ Aux Code | 15.94 | 5.14 | **9.04** | **5.99** |
| ShareGPT | Off-JCG | **22.18** | 6.01 | **8.79** | 5.89 |
| | + w/ Aux Code | 19.95 | 5.95 | 8.78 | **6.15** |
| WildChat | Off-JCG | **20.97** | **5.81** | 5.13 | 5.35 |
| | + w/ Aux Code | 16.65 | 4.81 | **8.91** | **5.90** |

We compared the training trajectories of our standard Off-JCG method against this version. The final performance comparison is shown in Table 2. Interestingly, while the introduction of auxiliary Judge Code did not lead to a higher peak performance compared to our standard method, it significantly altered the training dynamics.

As detailed in Appendix G, the standard Off-JCG method typically reached its optimal performance earlier in the training process. In contrast, the model trained with the auxiliary Judge Code exhibited a more prolonged learning phase, achieving its peak performance at a later stage. This suggests that the auxiliary Judge Code introduces additional complexity into the reward landscape, which slows down initial convergence but could potentially lead to a more thorough exploration of the policy space. While not yielding superior final results in our current setup, this finding opens up a promising avenue for future research into how structured "code ensembles" can be used to fine-tune the learning process.

### 4.5 QUALITATIVE CASE STUDIES

We present detailed case studies in the appendix. These examples highlight both the mechanics of our method and the resulting improvements in model capabilities.

- **Code Case Study (Appendix H):** We provide concrete examples of our programmatic rubrics. This includes the ***core*** Judge Code, which is generated from the universal prompt used throughout our main experiments (On-JCG and Off-JCG), and the ***auxiliary*** Judge Code, which was introduced specifically for the Code Scaling Study in Section 4.4.
- **Response Case Study (Appendix I):** We present a side-by-side comparison of responses generated by our Off-JCG versus the original base model.

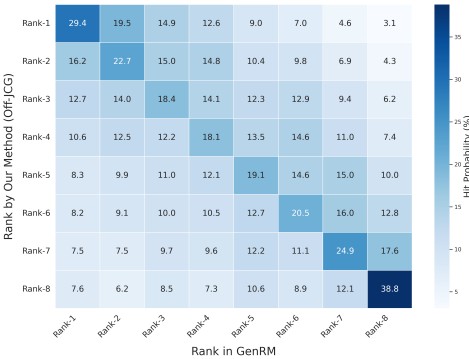

Figure 8: Rank Correspondence Heatmap between Judge Code Rewards and GenRM. Each cell (row $i$, column $j$) shows the probability that a response ranked $i$ by our method is ranked $j$ by the GenRM baseline.

## 5 ANALYSIS

### 5.1 THE NATURE OF JUDGE CODE REWARDS: A DIFFERENTIATED SUPERVISOR

The rewards generated by Judge Code are significantly correlated with, but not identical to, those of a strong teacher model (GenRM), positioning it as a differentiated supervisor (see Figure 8).

**Correlated, but Not Identical.** The heatmap reveals a clear positive correlation; for instance, a response ranked Rank-1 by our method has a 28.5% chance of also being ranked Rank-1 by GenRM, a value far exceeding the 12.5% expected from random chance. However, this imperfect agreement indicates that Judge Code does not simply replicate the teacher's preferences. Instead, it applies a set of explicit, programmatic rules, which structurally differs from GenRM's holistic evaluation process.

**An Effective "Weak" Supervisor.** Although its Top-1 precision against the GenRM is modest, Judge Code reliably distinguishes between high- and low-quality candidates. For example, a response ranked Rank-1 by our method has a 48.2% probability of being in the GenRM's Top-2, whereas a Rank-8 response has only a 3.1% chance of being a GenRM Top-1. This strong penalization of low-quality responses provides a stable and correct directional signal for RL training.

## 5.2 THEORETICAL PERSPECTIVE: PARTIAL REWARD AS AN UNBIASED GRADIENT ESTIMATOR

We provide a theoretical justification for the efficacy of our method by proving that the policy gradient derived from partial rewards is, in expectation, identical to the gradient from complete rewards. This frames our approach as a valid form of stochastic optimization. In this context, a "sample" $i$ refers to a prompt-response pair, and a "group" $g$ consists of all samples sharing the same prompt.

Our argument begins with the reward signals. Let the full reward be $R_{\text{full}}(i) = \frac{1}{N} \sum_{k=1}^{N} c_k(i)$, where $c_k(i) \in \{0, 1\}$ is an indicator for satisfying constraint $k$. The partial reward, $R_{\text{partial}}(i; k) = c_k(i)$, results from evaluating a single constraint $k$, uniformly sampled from the $N$ available constraints, i.e., with probability $P(k) = 1/N$. Crucially, the partial reward is a perfect unbiased estimator of the full reward, as its expectation over the random choice of $k$ is precisely the full reward:

$$\mathbb{E}_k[R_{\text{partial}}(i)] = \sum_{k=1}^{N} P(k) \cdot c_k(i) = \sum_{k=1}^{N} \frac{1}{N} c_k(i) = R_{\text{full}}(i) \tag{7}$$

This unbiased property extends directly to the GRPO advantage. For theoretical clarity, we analyze the unnormalized advantage $\hat{A}(i) = R(i) - \mu_g$, where $\mu_g$ is the group's mean reward. The resulting advantage estimator is also unbiased, meaning the expected partial advantage equals the deterministic full advantage. This holds because the group baseline $\mu_g$ is also estimated without bias (see Appendix D for details). Thus, for any sample $i$:

$$\mathbb{E}_k[\hat{A}_{\text{partial}}(i)] = \hat{A}_{\text{full}}(i) \tag{8}$$

This result directly implies that the policy gradient is unbiased. The gradient for a sample $i$ is proportional to $\nabla_\theta \log \pi_\theta(a_i|s_i) \cdot \hat{A}(i)$. Since the policy term is independent of the reward sampling, the expected gradient under partial rewards is identical to the full-reward gradient:

$$\mathbb{E}_k\left[ \nabla_\theta \log \pi_\theta(a_i|s_i) \cdot \hat{A}_{\text{partial}}(i) \right] = \nabla_\theta \log \pi_\theta(a_i|s_i) \cdot \mathbb{E}_k[\hat{A}_{\text{partial}}(i)]$$

$$= \nabla_\theta \log \pi_\theta(a_i|s_i) \cdot \hat{A}_{\text{full}}(i) \tag{9}$$

Therefore, while any individual training step is noisy, the unbiasedness of the gradient guarantees that the optimization trajectory is correct in expectation. The standard deviation normalization acts as an adaptive learning rate but does not alter this fundamental directional correctness.

## 6 CONCLUSION

In this work, we chart a new path for open-domain alignment, addressing the significant challenge of reward modeling on non-verifiable domains by demonstrating that directly guiding RL with programmatic rubrics instantiated as Judge Code is a viable and promising direction. We first established that partial rewards are sufficient for effective RL training, and then demonstrated that "compiling" evaluation criteria into executable **Judge Code** is a viable, low-cost paradigm. Moreover, we introduce the **JC-RL** framework, whose **Off-JCG** mode leverages reusable (`Query`, `Judge Code`) assets to achieve competitive performance more than 2x faster than strong GenRM baselines. This approach not only provides a practical solution but also opens up significant future research avenues in a more sophisticated code synthesis and enhanced alignment interpretability.

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

## A  THE USE OF LARGE LANGUAGE MODELS (LLMS)

During the preparation of this manuscript, we employed a Large Language Model (LLM) as a writing assistant. The use of the LLM was strictly limited to proofreading for grammatical errors and rephrasing sentences for improved clarity. The core research, methodology, and data analysis were conducted entirely by the authors.

## B  RELATED WORK

**RL with Verifiable Rewards.**  In verifiable domains like mathematics, programming, and games, progress is driven by Reinforcement Learning with Verifiable Rewards (RLVR), which leverages rule-based, automated feedback. This is exemplified in mathematics, where feedback is a pass/fail signal against a ground-truth solution (Albalak et al., 2025; Chen et al., 2025c); in programming, where rewards are tied to passing unit tests (Guo et al., 2025a); and in games, where the engine itself provides the outcome (Liu et al., 2025a; Chen et al., 2025a). The success of this RLVR paradigm stems from a shared principle: these domains are difficult to solve but easy to verify, satisfying the need for rapid, automated ground-truth evaluation (Guo et al., 2025a; He et al., 2025).

**RL with LLM-based Rewards.**  For non-verifiable tasks, the **LLM-as-a-Judge** paradigm marks a pivotal shift, relying on another powerful LLM for evaluation (Zheng et al., 2023). Initially centered on Reasoning Reward Models that generate a rationale before judging (Li et al., 2023; Ankner et al., 2024; Chen et al., 2025b; Guo et al., 2025b; Hong et al., 2025b; Liu et al., 2025d), this was refined by rubric-based methods that anchor evaluation in explicit criteria to ensure consistency (Gunjal et al., 2025; Huang et al., 2025b; Viswanathan et al., 2025; Lu, 2025; Zeng et al., 2025). The paradigm further evolves into co-evolving systems, where policy and reward models improve together through strategies like self-rewarding, where a model critiques its own output (Yuan et al., 2024; Team et al., 2025; Xiong et al., 2025; Zhang et al., 2025c), and co-optimization of distinct policy and reward models (Wang et al., 2025; Hong et al., 2025a).

**RL with Proxy Rewards.**  To bypass human annotation and achieve scalability, a third direction seeks to replace costly supervision with automatically generated proxy rewards. One family of approaches looks inward, leveraging an LLM's own internal state to derive these proxies. This includes methods based on output consistency via voting (Zhang et al., 2025b; Zuo et al., 2025; Liu et al., 2025b), internal confidence scores like token probabilities (Gao et al., 2025; Yu et al., 2025b; Kiruluta et al., 2025; Prabhudesai et al., 2025), or self-generated knowledge (Yuan et al., 2024; Yang et al., 2025; Zhao et al., 2025; Huang et al., 2025a). In contrast, other approaches look outward, relying on external automated sources such as simple output-property heuristics (Guo et al., 2025a) or data-centric signals from unlabeled corpora (Dong et al., 2025; Zweiger et al., 2025). While highly scalable, these unsupervised proxies are often simplistic and prone to reward hacking or model collapse (Liu et al., 2025c), highlighting the challenge of creating robust reward signals without direct verification or human oversight.

Our work challenges the conventional separation between these domains by pioneering the extension of RLVR to open-domain tasks. We achieve this by employing sample-specific "Judge Code" as a structured and programmatic form of proxy reward, which provides a scalable, RM-free alternative to traditional LLM-based rewards. This methodology unifies the efficiency of RLVR with the complex evaluation needs of non-verifiable domains, overcoming the fragility of simpler unsupervised proxies.

## C PROMPT TEMPLATE

User: The user asks a question, and the Assistant solves it. The assistant first thinks about the reasoning process in the mind and then provides the user with the final answer. The reasoning process and answer are enclosed within <think> </think> and <answer> </answer> tags, respectively, i.e., <think> reasoning process here </think><answer> answer here </answer>. Now the user asks you to solve a reasoning problem. After thinking, when you finally reach a conclusion, clearly state the solution within <answer> </answer> tags. Write the only correct answer that meets the instructions. User: You are a helpful assistant. {question} Assistant: <think>

Figure 9: The prompt with a Chain-of-Thought (CoT) template used in training contains a "question" placeholder, which will be replaced with a specific question.

## D DERIVATION OF THE UNBIASED ADVANTAGE ESTIMATOR

This appendix provides the detailed derivation showing that the partial advantage, $\hat{A}_{\text{partial}}(i)$, is an unbiased estimator of the full advantage, $\hat{A}_{\text{full}}(i)$.

The goal is to show that $\mathbb{E}_k[\hat{A}_{\text{partial}}(i)] = \hat{A}_{\text{full}}(i)$. We expand the term using the linearity of expectation:

$$\mathbb{E}_k[\hat{A}_{\text{partial}}(i)] = \mathbb{E}_k[R_{\text{partial}}(i) - \mu_{g,\text{partial}}] = \mathbb{E}_k[R_{\text{partial}}(i)] - \mathbb{E}_k[\mu_{g,\text{partial}}] \tag{10}$$

The first term, the expected partial reward for a fixed sample $i$, is:

$$\mathbb{E}_k[R_{\text{partial}}(i)] = \sum_{k=1}^{N} P(k) \cdot c_k(i) = \sum_{k=1}^{N} \frac{1}{N} c_k(i) = R_{\text{full}}(i) \tag{11}$$

The second term is the expected group baseline, where the expectation is taken over the independent random choices for all samples $j$ in the group $\mathcal{I}_g$:

$$\mathbb{E}_k[\mu_{g,\text{partial}}] = \mathbb{E}_k\left[ \frac{1}{|\mathcal{I}_g|} \sum_{j \in \mathcal{I}_g} R_{\text{partial}}(j) \right]$$

$$= \frac{1}{|\mathcal{I}_g|} \sum_{j \in \mathcal{I}_g} \mathbb{E}_k[R_{\text{partial}}(j)] \quad \text{(by linearity of expectation)}$$

$$= \frac{1}{|\mathcal{I}_g|} \sum_{j \in \mathcal{I}_g} R_{\text{full}}(j) \quad \text{(using Eq. 11)}$$

$$= \mu_{g,\text{full}} \tag{12}$$

Substituting the results from Eq. 11 and Eq. 12 back into Eq. 10, we confirm the advantage estimator is unbiased:

$$\mathbb{E}_k[\hat{A}_{\text{partial}}(i)] = R_{\text{full}}(i) - \mu_{g,\text{full}} = \hat{A}_{\text{full}}(i) \tag{13}$$

## E INDUCTION PROMPT

The design of induction prompts is predicated on the principle that structured, explicit reasoning must precede code generation to produce effective, sample-specific programmatic rubrics. The two-part structure serves as programmatic Chain-of-Thought (CoT), forcing the JCG to deconstruct the task and formulate a coherent evaluation plan before implementation.

**Core Induction Prompt (Figure 10)** This is the primary prompt used in our main experiments. Its role is to synthesize a "gain-focused" rubric that rewards desirable qualities in a response. It achieves this through three key instructions:

1. **Task Deconstruction:** It forces the JCG to analyze the input query ("processed question") to ground the evaluation criteria in the specific requirements of the user's request.
2. **Reference Analysis:** By requiring an analysis of provided responses ("contents"), it compels the JCG to calibrate its scoring logic and enhance its discriminating power by observing concrete examples of varying quality.

**Auxiliary Induction Prompt (Figure 11)**   In contrast, the auxiliary prompt, used in our code scaling study (Section 4.4), adopts an adversarial, "penalty-focused" stance. It instructs the JCG to act as a Quality Assurance (QA) analyst, focusing exclusively on identifying and penalizing common failure modes such as verbosity, irrelevance, or subtle hallucinations that might not violate hard constraints but degrade user experience. This provides a complementary signal, targeting defects rather than rewarding merits, allowing us to explore more complex reward landscapes.

You are an expert AI programmer and analyst. Your mission is to first analyze a task and then write a well-commented Python script to programmatically evaluate the quality of a generated text based on that analysis.
### **Your Core Mission**
    Your goal is to create a **bespoke Python scoring function** that is **hyper-specific** to the task defined below. It must be able to **meaningfully differentiate** the quality of the provided reference examples. A generic, one-size-fits-all script is considered a complete failure.
### **Context for This Specific Task**
    **1. The Task Instruction (`processed_question`):** This is the problem you need to solve. Your entire evaluation logic must revolve around its specific rules.
    ```
    {question}
    ```
    **2. Based on the above tasks, I have obtained {count} alternative builds, you need to use these builds as reference. Your analysis and code must be able to assign them a discriminating range of scores. The generated content to evaluate is:
    ```
    {contents}
    ```
### **Your Required Output Format**
Your output MUST be structured in the following two parts:
---
**Part 1: Evaluation Strategy Analysis**
    Before presenting the code, you must first provide a step-by-step breakdown of your evaluation plan. Explain your reasoning clearly. This section should cover:
    1. **Task Deconstruction & Core Criteria:** What are the key, non-negotiable requirements of the `Task Instruction`? (e.g., specific format, keywords, constraints).
    **Identify Core Criteria:** Based on the task type, break down the requirements into two categories-**Hard Criteria (Non-Negotiable):** and **Soft Criteria (Qualitative Measures):**. You need to clarify the judgment required for the task. Hard Criteria are binary, pass/fail rules. Violating them should lead to a very low score. These are nuanced, quality-based aspects that allow for a spectrum of scores. Your goal is to find **programmatic proxies** for these qualities.
    2. **Reference Analysis:** What patterns, keywords, or structures did you observe in the high-quality `Reference Examples`? How will you use these to differentiate scores?
    3. **Scoring Rubric:** Describe your proposed scoring logic. Explain how you will assign points. For example: "The base score is 5.0. It gets +2 points for correct JSON format, +1 for including the keyword 'summary', and -3 for exceeding 200 words." Justify your weighting.
    ---
**Part 2: Fully Commented Python Code**
    After the analysis, provide the final Python code.
**Code Requirements:**
    1. **Markdown Block:** The code MUST be enclosed in a single Python markdown block (```python ... ```).
    2. **Well-Commented:** The code MUST be thoroughly commented. Explain the purpose of the `compute_score` function, key variables, and complex logic blocks. Your comments should make the code easy for a human to understand.
    3. **Function Signature:** Must contain `def compute_score(original_prompt: str, generated_content: str) -> float:`.
    4. **Score Range:** Must return a `float` between **1.0 (worst)** and **10.0 (best)**.
    5. **CRITICAL - Task-Specific & Discriminating:** The logic must directly implement the strategy you outlined in Part 1. It must be custom-built for this task and have strong discriminating power.
    6. **Self-Contained:** Use only standard Python libraries (e.g., `re`, `json`). No external downloads allowed.
    ---
**Restrictions**
    1. **Hard requirements:** If the question mentions hard and predictable requirements, such as a word limit or a summary, you must emphasize them and return the lowest score if the mandatory restrictions are not met.
    2. **Internal comparison:** You need to distinguish between the passages I provided, so the code is also for them.
### **Final Instruction**
    You are now generating the analysis and custom evaluation code for the task: **"{processed_question}"**. Follow the two-part output format precisely.

Figure 10: Core induction prompt.

You are a senior AI Quality Assurance (QA) Analyst and a meticulous Python programmer. Your primary function is to anticipate and programmatically detect potential failure modes in an AI-generated text. Your goal is not to assess correctness, but to **identify and penalize common, yet subtle, quality defects.**

### **Your Core Mission**
Your goal is to create a **bespoke Python penalty-scoring function** that is **hyper-specific** to the *potential flaws* of the task defined below. It must be able to **programmatically flag** responses that are technically "correct" but practically unhelpful due to issues like verbosity, irrelevance, or subtle hallucinations. A generic penalty script is a complete failure.

### **Context for This Specific Task**
**1. The Task Instruction (`processed_question`):** This is the problem the AI was supposed to solve.
```
{processed_question}
```
**2. Sample Generated Text (`generated_content`):** Analyze those {count} example to brainstorm potential failure modes.
```
{contents}
```

### **Your Required Output Format**
Your output MUST be structured in the following two parts:

---
**Part 1: Defect Strategy Analysis**
Before presenting the code, you must first provide a step-by-step breakdown of your defect detection plan. Explain your reasoning as a QA analyst.
1. **Task-Specific Failure Mode Analysis:**
* **Deconstruct the Task:** What is the user's *true intent*? Is it a request for a quick fact, a creative piece, a detailed explanation, or a code snippet?
* **Anticipate "Lazy" or "Clever but Wrong" Paths:** For this specific task, what is the easiest way for an AI to produce a response that *looks* plausible but is low-quality?
* *Example for a simple fact question:* "The AI might pad the answer with generic, unnecessary historical context to appear more knowledgeable, which is a form of verbosity."
* *Example for a coding question:* "The AI might provide a working but overly complex or inefficient solution, or fail to include necessary comments and explanations."
* **Identify Programmatic "Red Flags":** Based on your analysis, what are the specific, code-detectable signals (red flags) that correlate with these failure modes? These are your **Penalty Criteria**.
2. **Penalty Rubric Design:**
* Describe your proposed penalty logic. Explain how you will assign penalty points for each red flag detected. A higher score means a WORSE response.
* **Justify Your Weighting:** Why is one defect more severe than another for *this specific task*?
* *Example:* "For a 'what is the capital' question, verbosity is a major flaw (penalty +5), while a minor, unrequested fact is a smaller issue (penalty +2). For a creative writing task, the opposite might be true."
---
**Part 2: Fully Commented Python Penalty Code**
After the analysis, provide the final Python code.
**Code Requirements:**
1. **Markdown Block:** The code MUST be enclosed in a single Python markdown block (```python ... ```).
2. **Well-Commented:** The code MUST be thoroughly commented. Explain the purpose of the `compute_penalty_score` function and how each block of logic corresponds to the "Red Flags" you identified.
3. **Function Signature:** Must contain `def compute_penalty_score(original_prompt: str, generated_content: str) -> float:`.
4. **Score Range:** Must return a `float` representing the **penalty score**. **0.0 means no defects detected**. A higher score indicates a more flawed response. 10.0 points represents the highest penalty, score return by code must be between 0.0 and 10.0
5. **CRITICAL - Defect-Focused & Discriminating:** The logic must directly implement the penalty strategy you outlined in Part 1. It must be custom-built to penalize the specific failure modes of this task.
6. **Self-Contained:** Use only standard Python libraries (e.g., `re`, `json`).
---
**Final Instruction**
You are now generating the defect analysis and penalty-scoring code for the task: **"{processed_question}"**.
Follow the two-part output format precisely. Your analysis must demonstrate deep, task-specific reasoning.

Figure 11: Auxiliary induction prompt.

# F TRAINING EFFICIENCY COMPARISON BETWEEN OFF-JCG AND GENRM.

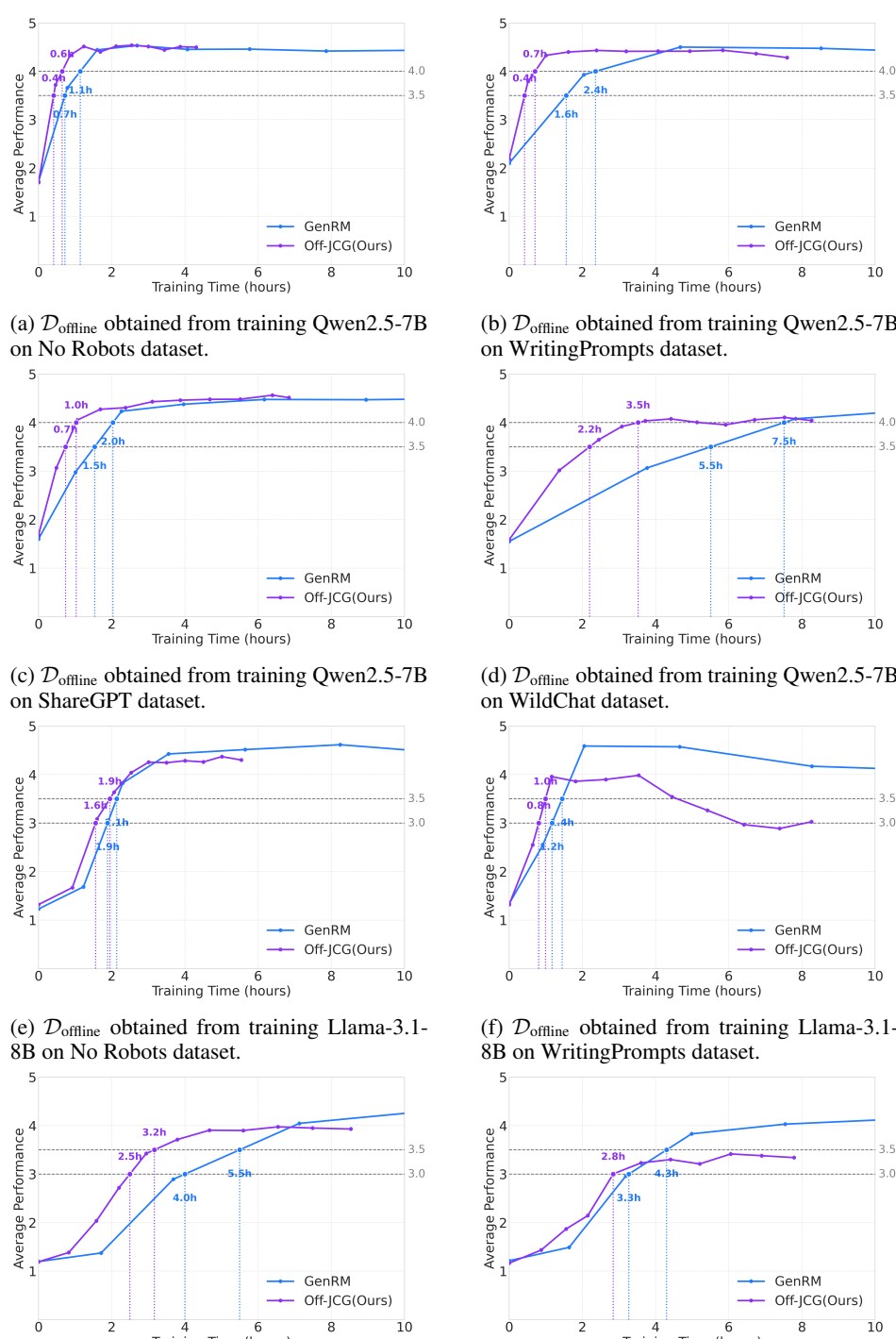

(a) $\mathcal{D}_{\text{offline}}$ obtained from training Qwen2.5-7B on No Robots dataset.

(b) $\mathcal{D}_{\text{offline}}$ obtained from training Qwen2.5-7B on WritingPrompts dataset.

(c) $\mathcal{D}_{\text{offline}}$ obtained from training Qwen2.5-7B on ShareGPT dataset.

(d) $\mathcal{D}_{\text{offline}}$ obtained from training Qwen2.5-7B on WildChat dataset.

(e) $\mathcal{D}_{\text{offline}}$ obtained from training Llama-3.1-8B on No Robots dataset.

(f) $\mathcal{D}_{\text{offline}}$ obtained from training Llama-3.1-8B on WritingPrompts dataset.

(g) $\mathcal{D}_{\text{offline}}$ obtained from training Llama-3.1-8B on ShareGPT dataset.

(h) $\mathcal{D}_{\text{offline}}$ obtained from training Llama-3.1-8B on WildChat dataset.

Figure 12: Comparison of the efficiency of Off-JCG and GenRM on four non-verifiable domain tasks using Qwen2.5-7B and Llama-3.1-8B.

# G TRAINING DETAILS

Table 3: Performance Evaluation of Qwen2.5-7B training with only **Core Judge Code** at Various Training Steps via On-JCG.

| Training set | Benchmark | Training steps | | | | |
|---|---|---|---|---|---|---|
| | | 20 steps | 40 steps | 60 steps | 80 steps | 100 steps |
| Aug-IFEval | IFEval | 40.67 | 55.08 | 64.14 | 69.32 | **70.61** |
| WritingPrompts | Creative Writing v3 | 67.34 | 68.15 | 70.45 | 69.42 | **71.11** |
| No Robots | AlpacaEval 2.0 | 15.51 | 29.18 | 28.91 | **31.05** | 29.18 |
| | MT-bench | 3.43 | 5.82 | 5.66 | **6.13** | 5.45 |
| ShareGPT | AlpacaEval 2.0 | 19.44 | 22.25 | **23.44** | 14.45 | 13.92 |
| | MT-bench | 5.12 | **5.87** | 4.97 | 4.23 | 3.77 |
| WildChat | AlpacaEval 2.0 | 15.66 | **16.56** | 12.82 | 13.16 | 8.66 |
| | MT-bench | 5.55 | **5.87** | 4.98 | 4.55 | 3.26 |

Table 4: Performance Evaluation of Qwen2.5-7B training with only **Core Judge Code** at Various Training Steps via Off-JCG.

| Training set | Benchmark | Training steps | | | | |
|---|---|---|---|---|---|---|
| | | 20 steps | 40 steps | 60 steps | 80 steps | 100 steps |
| Aug-IFEval | IFEval | 42.33 | 56.75 | 64.70 | 66.73 | **69.69** |
| WritingPrompts | Creative Writing v3 | 64.84 | 67.92 | 68.43 | **70.05** | 68.59 |
| No Robots | AlpacaEval 2.0 | 16.99 | 23.59 | 27.27 | **29.67** | 28.89 |
| | MT-bench | 5.91 | 6.13 | **6.16** | 5.96 | 5.95 |
| ShareGPT | AlpacaEval 2.0 | 14.44 | **22.18** | 17.42 | 16.59 | 8.51 |
| | MT-bench | 5.89 | **6.01** | 4.92 | 3.04 | 1.39 |
| WildChat | AlpacaEval 2.0 | 16.43 | **20.97** | 20.17 | 12.05 | 8.41 |
| | MT-bench | 5.78 | **5.81** | 5.01 | 3.11 | 2.94 |

Table 5: Performance Evaluation of Llama-3.1-8B training with only **Core Judge Code** at Various Training Steps via On-JCG.

| Training set | Benchmark | Training steps | | | | |
|---|---|---|---|---|---|---|
| | | 20 steps | 40 steps | 60 steps | 80 steps | 100 steps |
| Aug-IFEval | IFEval | 40.67 | 62.29 | 65.80 | 70.61 | **72.64** |
| WritingPrompts | Creative Writing v3 | 41.24 | 49.86 | 56.79 | 62.35 | 63.22 |
| No Robots | AlpacaEval 2.0 | 6.17 | **13.35** | 11.22 | 9.57 | 5.54 |
| | MT-bench | 5.19 | **6.33** | 5.58 | 3.85 | 3.13 |
| ShareGPT | AlpacaEval 2.0 | 4.34 | **9.96** | 7.53 | 5.96 | 3.84 |
| | MT-bench | 5.01 | **6.22** | 4.76 | 4.22 | 3.35 |
| WildChat | AlpacaEval 2.0 | 2.35 | 5.23 | **6.01** | 4.98 | 4.02 |
| | MT-bench | 3.26 | **5.88** | 5.69 | 4.68 | 3.03 |

Table 6: Performance Evaluation of Llama-3.1-8B training with only **Core Judge Code** at Various Training Steps via Off-JCG.

| Training set | Benchmark | Training steps | | | | |
|---|---|---|---|---|---|---|
| | | 20 steps | 40 steps | 60 steps | 80 steps | 100 steps |
| Aug-IFEval | IFEval | 35.67 | 51.20 | 57.49 | 60.07 | **63.96** |
| WritingPrompts | Creative Writing v3 | 36.81 | 47.68 | 50.42 | 53.64 | **55.89** |
| No Robots | AlpacaEval 2.0 | 4.84 | 6.82 | 8.42 | 7.46 | **9.04** |
| | MT-bench | 5.91 | **6.13** | 6.16 | 6.00 | 5.95 |
| ShareGPT | AlpacaEval 2.0 | 6.14 | **8.79** | 5.32 | 4.24 | 3.25 |
| | MT-bench | 5.66 | **5.89** | 4.92 | 3.04 | 1.39 |
| WildChat | AlpacaEval 2.0 | **5.13** | 1.67 | 1.14 | 1.09 | 0.66 |
| | MT-bench | **5.35** | 2.39 | 1.03 | 0.88 | 0.56 |

Table 7: Performance Evaluation of Qwen2.5-7B training with both **Core** and **Auxiliary Judge Code** at Various Training Steps via On-JCG.

| Training set | Benchmark | Training steps | | | | |
|---|---|---|---|---|---|---|
| | | 20 steps | 40 steps | 60 steps | 80 steps | 100 steps |
| Aug-IFEval | IFEval | 40.67 | 55.08 | 64.14 | **69.32** | 68.02 |
| WritingPrompts | Creative Writing v3 | 67.34 | 68.15 | 70.45 | 69.42 | **71.11** |
| No Robots | AlpacaEval 2.0 | 10.67 | 14.76 | 17.11 | 16.38 | **21.14** |
| | MT-bench | 4.78 | 5.15 | 4.89 | 5.06 | **5.29** |
| ShareGPT | AlpacaEval 2.0 | 10.88 | 17.60 | 18.92 | 19.27 | **21.58** |
| | MT-bench | 5.16 | 5.63 | **5.73** | 5.64 | 5.60 |
| WildChat | AlpacaEval 2.0 | 7.61 | 13.00 | 14.20 | 14.78 | **17.88** |
| | MT-bench | 3.63 | 5.02 | 4.61 | **5.20** | 5.08 |

Table 8: Performance Evaluation of Qwen2.5-7B training with both **Core** and **Auxiliary Judge Code** at Various Training Steps via Off-JCG.

| Training set | Benchmark | Training steps | | | | |
|---|---|---|---|---|---|---|
| | | 20 steps | 40 steps | 60 steps | 80 steps | 100 steps |
| Aug-IFEval | IFEval | 42.33 | 56.75 | 64.70 | 66.73 | **69.69** |
| WritingPrompts | Creative Writing v3 | 67.42 | 68.22 | 69.45 | **71.21** | 70.29 |
| No Robots | AlpacaEval 2.0 | 13.49 | 14.83 | 15.08 | 14.61 | **15.94** |
| | MT-bench | 4.55 | 4.78 | 5.09 | **5.16** | 5.14 |
| ShareGPT | AlpacaEval 2.0 | 11.05 | 15.88 | 17.75 | 19.87 | **19.95** |
| | MT-bench | 4.24 | 5.38 | 5.39 | 5.61 | **5.95** |
| WildChat | AlpacaEval 2.0 | 13.85 | **16.70** | 12.83 | 14.92 | 16.65 |
| | MT-bench | 3.56 | 4.27 | 3.68 | 4.64 | **4.81** |

Table 9: Performance Evaluation of Llama-3.1-8B training with both **Core** and **Auxiliary Judge Code** at Various Training Steps via On-JCG.

| Training set | Benchmark | Training steps | | | | |
|---|---|---|---|---|---|---|
| | | 20 steps | 40 steps | 60 steps | 80 steps | 100 steps |
| Aug-IFEval | IFEval | 40.67 | 62.29 | 65.80 | 70.61 | **72.64** |
| WritingPrompts | Creative Writing v3 | 43.64 | 51.25 | 58.91 | 62.55 | **64.34** |
| No Robots | AlpacaEval 2.0 | 7.35 | 9.11 | 11.56 | 12.98 | **13.42** |
| | MT-bench | 3.84 | 4.05 | 4.21 | 4.45 | **4.59** |
| ShareGPT | AlpacaEval 2.0 | 4.47 | 6.88 | 8.92 | **10.53** | 10.15 |
| | MT-bench | 2.21 | 3.15 | 3.98 | 4.52 | **4.63** |
| WildChat | AlpacaEval 2.0 | 4.02 | 5.77 | 7.51 | 8.89 | **9.06** |
| | MT-bench | 2.11 | 3.05 | 3.90 | **4.42** | 4.42 |

Table 10: Performance Evaluation of Llama-3.1-8B training with both **Core** and **Auxiliary Judge Code** at Various Training Steps via Off-JCG.

| Training set | Benchmark | Training steps | | | | |
|---|---|---|---|---|---|---|
| | | 20 steps | 40 steps | 60 steps | 80 steps | 100 steps |
| Aug-IFEval | IFEval | 11.65 | 54.53 | 59.52 | 59.33 | **63.96** |
| WritingPrompts | Creative Writing v3 | 42.14 | 47.62 | 51.01 | 54.29 | **56.88** |
| No Robots | AlpacaEval 2.0 | 6.12 | 7.12 | 7.55 | 8.27 | **9.04** |
| | MT-bench | 3.58 | 4.11 | 4.40 | 4.45 | **5.99** |
| ShareGPT | AlpacaEval 2.0 | 4.77 | 7.58 | **9.11** | 8.78 | 7.82 |
| | MT-bench | 2.25 | 4.10 | 5.45 | **6.15** | 4.58 |
| WildChat | AlpacaEval 2.0 | 6.00 | 7.72 | **8.91** | 8.12 | 8.66 |
| | MT-bench | 2.12 | 4.92 | **5.90** | 4.75 | 4.32 |

This section provides the detailed, step-by-step performance data for all experiments, offering a granular view of the training dynamics. The tables are organized by model, training mode (On/Off-JCG), and the inclusion of auxiliary Judge Code. Tables 3–6 detail the performance of our standard framework, while Tables 7–10 correspond to the Code Scaling study (Section 4.4) where an auxiliary, defect-focused Judge Code was added. A deep analysis of these tables reveals a critical finding regarding the role of this auxiliary Judge Code.

**Analysis: Auxiliary Judge Code as a Training Stabilizer to Mitigate Performance Degradation.**
A primary and consistent finding from our Code Scaling study is the role of the auxiliary Judge Code in enhancing training stability and mitigating catastrophic performance degradation, especially on open-ended conversational datasets. While standard training often leads to rapid initial gains followed by a sharp decline, the addition of the auxiliary Judge Code acts as a regularizer, preventing the model from over-optimizing on the primary reward signal and thus preserving performance in later training stages. This effect is consistently observed across both models and training modes.

First, we examine the **Offline (Off-JCG)** training mode, which highlights the starkest contrast.

- For **Llama-3.1-8B** on `WildChat` (comparing Table 6 with Table 10), the standard method's performance peaks at **5.13** (20 steps) and then collapses almost entirely to **0.66**. In stark contrast, the version with auxiliary Judge Code peaks higher at **8.91** and remains remarkably resilient, ending at **8.66**.

- A similar dramatic stabilization occurs on `ShareGPT`. The standard Llama-3.1-8B's score plummets from a peak of **8.79** to **3.25**, whereas the auxiliary-trained model maintains its performance, decaying only slightly from **9.11** to **7.82**.

- The same trend holds for **Qwen2.5-7B** on `ShareGPT` (comparing Table 4 with Table 8). The standard method's score drops from **22.18** to **8.51**, while the auxiliary version shows a much more stable, prolonged learning curve, peaking at the final steps.

Next, we analyze the **Online (On-JCG)** training mode, which confirms the generality of this stabilizing effect.

- For **Qwen2.5-7B** on `WildChat` (comparing Table 3 with Table 7), the standard method's performance peaks at **16.56** (40 steps) and then rapidly degrades to **8.66**. The auxiliary Judge Code version, however, shows continuous improvement throughout, peaking at **17.88** at the final 100-step checkpoint, completely avoiding the performance collapse.
- This pattern is also evident with **Llama-3.1-8B** on `No Robots` (comparing Table 5 with Table 9). The standard model's score drops from a peak of **13.35** to **5.54**, while the auxiliary-trained model maintains a stable high-performance trajectory, peaking at **13.42**.

Taken together, these comprehensive results provide compelling evidence that the auxiliary Judge Code is not merely an add-on but a crucial component for robust training on complex, open-domain tasks. By penalizing common failure modes, it prevents the policy from overfitting to a narrow, "gain-focused" reward signal. This creates a more balanced and realistic reward landscape, leading to more generalizable policies that are resilient to performance degradation over longer training horizons.

## H  CODE CASE STUDY

This section provides concrete examples of our programmatic rubrics to illustrate the distinct roles of the **Core Judge Code** and the **Auxiliary Judge Code**. The core Judge Code is designed to reward desirable qualities ("gain-focused"), while the auxiliary Judge Code adopts an adversarial, "penalty-focused" stance.

### H.1  CASE 1: FACT-BASED SUMMARIZATION (FEDERAL RESERVE)

As shown in Figure 13 and 14. The evaluation for this fact-based task is precise and objective.

**Core Judge Code Logic.**  The core Judge Code acts as a "checklist-based grader". Its primary logic is to first verify the response's relevance to the topic and then to additively increase the score for each key factual point from the source text that is correctly included in the summary. It rewards factual completeness.

**Auxiliary Judge Code Logic.**  The auxiliary Judge Code functions as a "skeptical fact-checker". It focuses on penalizing common summarization errors, such as the inclusion of fabricated details (hallucinations), irrelevant information not present in the source text, or the omission of critical phrases.

### H.2  CASE 2: OPEN-ENDED CREATIVE WRITING (CHAINED TO THE FLOOR)

As shown in Figure 15 and 16. The evaluation for this creative task shifts from factual accuracy to narrative quality.

**Core Judge Code Logic.**  The core Judge Code serves as a "narrative quality heuristic". It first ensures the story adheres to the prompt's core elements (e.g., "chained", "priest"). It then rewards the presence of keywords that act as proxies for good storytelling, such as indicators of narrative coherence, emotional depth, and descriptive richness.

**Auxiliary Judge Code Logic.**  The auxiliary Judge Code acts as a "literary critic". Its logic is to penalize common pitfalls in creative writing, such as an over-reliance on clichés and abstract terms, jarring shifts in tense or perspective, and verbose, unstructured sentences.

"Original Question":
"What does the following article say about the Federal Reserve?\n\nThe Federal Reserve reinforced its fight against high inflation Wednesday by raising its key interest rate by a quarter-point to the highest level in 16 years. But the Fed also signaled that it may now pause the streak of 10 rate hikes that have made borrowing for consumers and businesses steadily more expensive.\n\nIn a statement after its latest policy meeting, the Fed removed a sentence from its previous statement that had said "some additional" rate hikes might be needed. It replaced it with language that said it will consider a range of factors in "determining the extent" to which future hikes might be needed.\n\nSpeaking at a news conference, Chair Jerome Powell said the Fed has yet to decide whether to suspend its rate hikes. But he pointed to the change in the statement's language as confirming at least that possibility.\n",

"Core Judge Code":

```python
import re
def compute_score(original_prompt: str, generated_content: str) -> float:
    score = 5.0

    # 1. Check relevance to Federal Reserve
    fed_keywords = ['Federal Reserve', 'Fed', 'interest rate', 'Jerome Powell', 'rate hikes']
    is_relevant = any(keyword.lower() in generated_content.lower() for keyword in fed_keywords)
    if not is_relevant:
        return 1.0
    score += 3.0

    # 2. Check inclusion of key points
    key_points = [
        'raised its key interest rate by a quarter-point',
        'pause the streak of 10 rate hikes',
        'removed a sentence from its previous statement',
        'Jerome Powell said the Fed has yet to decide'
    ]
    included_points = 0
    for point in key_points:
        if point.lower() in generated_content.lower():
            included_points += 1
    score += min(included_points, 4)

    # 3. Check for inaccuracies
    if 'sand turtles' in generated_content.lower() or 'color' in generated_content.lower():
        score -= 2.0

    # 4. Evaluate clarity
    sentences = re.split(r'[.!?]', generated_content)
    avg_sentence_length = sum(len(sentence.split()) for sentence in sentences) / max(1, len(sentences))
    if avg_sentence_length > 20:
        score -= 1.0
    else:
        score += 1.0

    # 5. Assess level of detail
    if included_points >= 2:
        score += 1.0
    if included_points >= 3:
        score += 2.0

    # 6. Check originality
    words = generated_content.lower().split()
    if len(words) > 0:
        unique_words = set(words)
        repetition_ratio = len(unique_words) / len(words)
        if repetition_ratio < 0.5:
            score -= 1.0
        else:
            score += 1.0
    score = max(1.0, min(10.0, score))

    return score
```

Figure 13: Core Judge Code case 1.

```
"Auxiliary Judge Code":
import re
def compute_penalty_score(original_prompt: str, generated_content: str) -> float:
    penalty = 0.0  # Initialize penalty score
    # Key phrases that must appear in quality summaries from original articles
    required_phrases = [
        "Federal Reserve",
        "interest rate",
        "quarter-point",
        "highest level in 16 years",
        "pause the streak of 10 rate hikes",
        "removed a sentence",
        "some additional",
        "determining the extent",
        "Jerome Powell"
    ]
    # Task-irrelevant phrases
    irrelevant_phrases = [
        "color",
        "wavelength",
        "light",
        "turtle",
        "Sahara",
        "celebrity",
        "Aristotle"
    ]
    # Fabricated phrases (not in original articles)
    fabricated_phrases = [
        "June 1st",
        "decided to suspend",
        "definitely pausing"
    ]
    # 1. Check irrelevant passages
    passages = re.split(r'<Passage id:\d+>', generated_content)
    passages = [p.strip() for p in passages if p.strip()]
    irrelevant_passages = 0
    for passage in passages:
        if any(irrelevant in passage.lower() for irrelevant in irrelevant_phrases):
            irrelevant_passages += 1
    penalty += irrelevant_passages * 3.0
    # 2. Check missing key information
    missing_info = 0
    for phrase in required_phrases:
        if phrase not in generated_content:
            missing_info += 1
    if missing_info > 0:
        penalty += 2.0
    # 3. Check hallucinated content
    hallucinations = 0
    for phrase in fabricated_phrases:
        if phrase in generated_content:
            hallucinations += 1
    penalty += hallucinations * 4.0
    # 4. Check redundancy (repeated passages)
    unique_passages = set()
    repeated_passages = 0
    for passage in passages:
        normalized = re.sub(r'[^\w\s]', '', passage.lower())
        if normalized in unique_passages:
            repeated_passages += 1
        else:
            unique_passages.add(normalized)
    penalty += repeated_passages * 1.0
    penalty = min(penalty, 10.0)

    return penalty
```

Figure 14: Auxiliary Judge Code case 1.

"Original Question":
"You wake up chained to stone floors . Your family stands next to a priest , sobbing . What happened ?\n",

"Core Judge Code":

```python
import re
def compute_score(original_prompt: str, generated_content: str) -> float:
    score = 5.0
    # 1. Check existence of key elements (hard criteria)
    key_elements = {
        'chained': r'chain(ed|ing|s)|strapped|bound|restrained',
        'family sobbing': r'family|mother|father|siblings|sobbing|crying|tears',
        'priest': r'priest|cleric|robes|chalice|divine'
    }
    missing_elements = []
    for element, pattern in key_elements.items():
        if not re.search(pattern, generated_content, re.IGNORECASE):
            missing_elements.append(element)
    if missing_elements:
        score -= 3.0
    # 2. Check narrative coherence (soft criteria)
    coherence_indicators = [
        r'because|since|as a result|therefore|thus',
        r'first|next|then|finally|after|before',
        r'however|but|although|despite'
    ]
    coherence_score = 0.0
    for indicator in coherence_indicators:
        if re.search(indicator, generated_content, re.IGNORECASE):
            coherence_score += 0.5
    score += min(coherence_score, 1.5)
    # 3. Check emotional engagement (soft criteria)
    emotional_indicators = [
        r'panic|fear|dread|terror|horror',
        r'sad|sorrow|grief|despair|anguish',
        r'confus(ed|ion)|bewilder(ed|ment)|disorient(ed|ation)'
    ]
    emotional_score = 0.0
    for indicator in emotional_indicators:
        if re.search(indicator, generated_content, re.IGNORECASE):
            emotional_score += 0.5
    score += min(emotional_score, 1.0)
    # 4. Check descriptive richness (soft criteria)
    descriptive_indicators = [
        r'cold|stone|iron|dungeon|dark|shadow',
        r'tears|sobbing|whispers|echo|voice',
        r'robe|chalice|candlelight|priest|divine'
    ]
    descriptive_score = 0.0
    for indicator in descriptive_indicators:
        if re.search(indicator, generated_content, re.IGNORECASE):
            descriptive_score += 0.5
    score += min(descriptive_score, 1.0)
    # 5. Check originality (soft criteria)
    original_phrases = [
        r'lavender sachet|crumpled newspaper|lightning illustrations',
        r'hidden pocket|teapot|floorboard',
        r'ocean of emotions|tides of agony|cosmic sky'
    ]
    original_score = 0.0
    for phrase in original_phrases:
        if re.search(phrase, generated_content, re.IGNORECASE):
            original_score += 0.5
    score += min(original_score, 0.5)
    score = max(1.0, min(10.0, score))
    return round(score, 2)
```

Figure 15: Core Judge Code case 2.

```
"Auxiliary Judge Code":
import re
from typing import List, Dict
def compute_penalty_score(original_prompt: str, generated_content: str) -> float:
    penalty = 0.0
    passages = re.split(r'<Passage id:\d+>', generated_content)
    passages = [p.strip() for p in passages if p.strip()]
    # 1. Check irrelevance (contains task-unrelated keywords)
    IRRELEVANT_TERMS = [
        "peeing", "magical device", "city", "park", "LEDs", "disco-esque",
        "urban", "caves", "streamed down", "adventure", "oasis"
    ]
    irrelevant_count = sum(
        len(re.findall(rf"\b{term}\b", passage, re.IGNORECASE))
        for term in IRRELEVANT_TERMS
        for passage in passages
    )
    penalty += min(3.0, irrelevant_count * 0.5)
    # 2. Check over-abstraction (contains excessive abstract terms)
    ABSTRACT_TERMS = [
        "soul", "divine", "cosmic", "fate", "transformation", "allegorical",
        "metaphor", "spiritual", "eternal", "profound"
    ]
    total_words = sum(len(passage.split()) for passage in passages)
    abstract_count = sum(
        len(re.findall(rf"\b{term}\b", passage, re.IGNORECASE))
        for term in ABSTRACT_TERMS
        for passage in passages
    )
    if total_words > 0:
        abstract_ratio = abstract_count / total_words
        penalty += min(2.0, abstract_ratio * 100)
    # 3. Check redundancy/verbosity (excessively long sentences without logical progression)
    avg_sentence_lengths = []
    for passage in passages:
        sentences = re.split(r'[.!?]', passage)
        sentences = [s.strip() for s in sentences if s.strip()]
        if sentences:
            avg_len = sum(len(s.split()) for s in sentences) / len(sentences)
            avg_sentence_lengths.append(avg_len)
    if avg_sentence_lengths:
        overall_avg = sum(avg_sentence_lengths) / len(avg_sentence_lengths)
        if overall_avg > 15:
            penalty += min(1.5, (overall_avg - 15) * 0.1)
    # 4. Check incoherence (confusing shifts in pronouns/tenses)
    pronoun_shifts = 0
    tense_shifts = 0
    for passage in passages:
        if (" you " in passage.lower() and " i " in passage.lower()) or \
           (" you " in passage.lower() and " they " in passage.lower()):
            pronoun_shifts += 1
        past_verbs = len(re.findall(r"\b(was|were|had|did)\b", passage.lower()))
        present_verbs = len(re.findall(r"\b(am|is|are|have|do)\b", passage.lower()))
        if past_verbs > 2 and present_verbs > 2:
            tense_shifts += 1
    penalty += min(2.0, (pronoun_shifts + tense_shifts) * 0.5)
    # 5. Check missing core elements (lacks key content required by the task)
    core_elements = ["family", "priest", "chain", "sobbing", "stone floor"]
    element_counts = {elem: 0 for elem in core_elements}
    for passage in passages:
        for elem in core_elements:
            element_counts[elem] += len(re.findall(rf"\b{elem}\b", passage, re.IGNORECASE))
    total_elements = sum(element_counts.values())
    if total_elements < 3:
        penalty += min(1.5, (3 - total_elements) * 0.5)
    return min(10.0, round(penalty, 1))
```

Figure 16: Auxiliary Judge Code case 2.

# I    RESPONSE CASE STUDY

"Original Question":
Make a vividly detailed and comedic fictional Zoey 101 story about Chase Matthews and Logan Reese (Chase's crush) relaxing in pond au naturel (compliment Logan's face, butt and physique). Logan puts on a relieved grin as the water behind him bubbles like he farted. a few fish die from the awful stench of the fart underwater. Logan begs Chase to not tell anyone, and Chase keeps it a secret due to how much he loves Logan. Add dialogue throughout the story

"Base Model Response":
### Assistant: <think> How can I create a hilarious story between Chase Matthews and Logan Reese, focusing on a humorous pond experience? I'll emphasize Logan's attractive features and create a funny fart-related incident. </think>
<answer>
In the beautiful, hilly town of Positano, Italy, there's a local pond, a hidden gem for secluded relaxation. On a blissful morning, Chase Matthews and Logan Reese found themselves drawn to the pond, deciding to spend their afternoon au naturel in the picturesque setting.

Chase, knowing that he has a special affection for Logan, was nervous and excited about sharing this unique moment with his crush. After stripping down to their swimming trunks, they settled at the pond's edge, admiring the ducks gliding gracefully on the water.

"Logan, you look absolutely stunning here," Chase remarked, his eyes never leaving Logan's face. "You always do."
Logan's face lit up, the sun reflecting off his sweaty, tousled hair and highlighting his chiseled features. He smiled, knowing Chase's words resonated with genuine admiration. "You don't look too bad yourself, Chase. Nice physique."
As the afternoon progressed and the sun began to sparkle on the pond's surface, Logan started to feel a rumble. "Chase, I have to tell you something," he said, looking slightly apprehensive.
"I'm here to listen, Logan. You can tell me anything," Chase replied, his heart fluttering with concern.
"Yeah, so here it goes. My bladder is just about to overfill. Would you mind if we take a quick break?"
Chase, understanding Logan's hectic schedule and knowing the stress behind the migration-induced bladder issues, agreed with a reassuring grin. "Of course not, Logan. Just wish I knew you wanted to take care of your bladder a bit earlier."
As they stood up, Logan's movements were not quite in sync, causing him to accidentally bend his knees too far, resulting in... a powerful underwater fart. A whirlwind of bubbles surfaced underneath him, and before he could comprehend what was happening, a few unfortunate fish began to float lifelessly to the surface.
Logan, horrified, blubbered, "Chase, please don't tell anyone."
Chase, being the supportive and empathetic friend, placed his arms around Logan and whispered softly, "I won't, Logan. I promise it's not your fault anyway. I mean, they know by now that Positano has lots of fishy fruits."
Despite the initial shock and dismay, they spent the rest of the afternoon enjoying the serene pond to its fullest, making up the questionable morning with its beautiful surroundings and chasing butterflies together. The story of the awkward underwater fart in Positano was etched in Venetian tales, remembered as a humorous episode in their friendship - surprisingly the topic of no conversation throughout their years at CP.
As far as Chase and Logan is concerned, it was nothing but a delightful, silly anecdote that highlights their journey of acceptance and friendship.
</answer>

"Off-JCG Response":
### Assistant: <think> I will create a comedic Zoey 101 story featuring Chase Matthews and Logan Reese relaxing in a pond au naturel with vivid details and witty dialogue.</think>
<answer>
In a whimsical twist of Zoey 101, Chase Matthews and Logan Reese found themselves relaxing in a pond au naturel. Chase's crush, Logan, had the most captivating face, an enviable butt, and a well-defined physique that made Chase's heart skip a beat.
As they floated together near the pond's edge, Logan put on a relieved grin as the water behind him bubbled like he farted.
Chase couldn't help but burst into laughter, commenting, "Logan, you're killing me with that relieved grin! Is that a newfound relaxation technique?"
Logan playfully fished for an excuse, "Ah, it's just the water calming me down. I'm not really aware of anything else." A few fish swimming nearby were overcome by the awful stench of the fart underwater and sadly died.
Logan's eyes widened in panic, "Chase, please don't tell anyone! I'll keep it between us, I promise."
Chase kept it a secret, not just due to Logan's love for him, but also because he couldn't resist the absurdity of the situation. "Don't worry, Logan, your little water-induced fart will remain our secret. You know I love you too much to spill the beans."
Logan let out a big sigh of relief, "Thank you, Chase! I couldn't have done it without you."
Chase chuckled, "Anytime, Logan. Let's focus on enjoying our time in paradise. Cheating death by fish stench is worth the laugh."
As they continued floating in the pond, Chase couldn't help but share a glance at Logan's face, butt, and physique, savoring the moment and knowing that their secret would remain forever between them.
</answer>

Figure 17: The response of Off-JCG model precisely recreated the user's specified core plot (the fart killing the fish) and character motivation (keeping the secret out of love), faithfully adhering to all detailed instructions. In contrast, the base model deviated significantly by inventing a new setting and illogical reasons, ultimately failing to deliver on the user's central request.

These case studies, spanning both creative and fact-based tasks, collectively illustrate the performance improvements achieved through our JC-RL framework. In the creative writing task (Figure 17), the Off-JCG model demonstrates a superior capacity for adhering to a complex and unusual

"Original Question":
How much has the City of Los Angeles spent fighting dust pollution from Owens Lake?

Once a lake bed is exposed, winds kick up ferocious dust storms. Those windblown sediments contribute to air pollution and can contribute to asthma, lung cancer and cardiopulmonary disease, among other health issues. Owens Lake has been among the largest sources of dust pollution in the nation. The City of Los Angeles has spent more than $2.5 billion mitigating the dust through projects at the lake bed such as shallow flooding, seeding and planting vegetation, spreading gravel or tilling the ground.

"Base Model Response":
### Assistant: <thinking> Processing the information given to determine the amount spent on fighting dust pollution.</thinking>
<answer> $2.5 billion </answer>

"Off-JCG Response":
### Assistant: <think>
To determine how much the City of Los Angeles has spent fighting dust pollution from Owens Lake, we need to analyze the given information and extract the relevant financial details. We should look for specific amounts or ranges mentioned in the text related to the spending for dust mitigation projects at the lake.
</think>
<answer>
Based on the information provided, the City of Los Angeles has spent more than $2.5 billion mitigating the dust pollution from Owens Lake through various projects. These projects include shallow flooding, seeding and planting vegetation, spreading gravel, and tilling the ground. The exact amount spent may be more than $2.5 billion, but the precise figure is not explicitly stated in the given text.
</answer>

Figure 18: The response of Off-JCG model is more precise by capturing the key qualifier "more than" for a complete answer, whereas the base model's simple numerical extraction is inaccurate.

set of plot and character constraints, whereas the base model defaults to a more generic, "safer" narrative. Similarly, in the factual extraction task (Figure 18), the Off-JCG model learns to capture critical linguistic subtleties like the qualifier "more than", moving beyond simple keyword matching to a more precise and faithful representation of the source information. This demonstrates that training with programmatic, partial rewards effectively guides the model to achieve better performance across diverse tasks.

