# OpenReview forum: "All in RLVR on Non-Verifiable Domains"
_ICLR.cc/2026/Conference — ICLR 2026 Conference Withdrawn Submission_

### Official Review · Reviewer_vjxV · 2025-10-27

**Soundness:** 3
**Presentation:** 3
**Contribution:** 2
**Rating:** 4
**Confidence:** 5

**Summary:**

This paper challenges the traditional view that Reinforcement Learning with Verifiable Rewards (RLVR) only applies to verifiable domains and proposes a method to extend it to non-verifiable open domains. First, a pilot study confirms that sample-level partial and imperfect rewards can effectively guide RL training when the dataset has sufficient coverage and diversity. Then, it introduces a Judge Code Generator (JCG) to produce sample-specific Judge Code, replacing traditional reward models, and builds the Judge Code-guided Reinforcement Learning (JC-RL) framework with two modes: On-JCG (generating data in real time) and Off-JCG (using pre-generated data). Experiments show that Off-JCG matches the performance of the generative reward model (GenRM) in non-verifiable domains while achieving over 2x faster training speed. It also verifies the cross-model and cross-scale generalization of the offline dataset.

**Strengths:**

1. This paper breaks the inherent notion that RLVR is limited to verifiable domains, proposes using sample-specific Judge Code to solve reward design problems in non-verifiable domains, opens up a new direction for RLVR applications, and has significant theoretical and practical significance.

2. The Off-JCG mode skips the Judge Code generation step, achieving over 2x faster training speed than GenRM while maintaining comparable performance. It balances efficiency and effectiveness, reducing RL training costs in non-verifiable domains.

3. The offline dataset (Query, Judge Code) can be transferred across different model architectures (e.g., Qwen2.5-7B and Llama-3.1-8B) and model scales (from 7B to 14B, 32B), proving its universality and scalability and enhancing the practical value of the method.

**Weaknesses:**

1. Although the title "All in RLVR on Non-Verifiable Domains" is intriguing, its contribution seems somewhat overstated. There is a discrepancy between the actual content and the generally recognized concept of "Non-Verifiable Domains". The evaluated benchmarks include the Creative Writing Benchmark v3, AlpacaEval 2.0, and MT-Bench. As a technique originating from the reasoning field, I am more concerned about whether this method can be extended to the verification of reasoning in fields such as mathematical proofs, physics, chemistry, and biology.

2. The paper mentions that "The Off-JCG mode skips the Judge Code generation step, achieving over 2x faster training speed than GenRM while maintaining comparable performance." However, I cannot find the prompts for GenRM as a verifier, nor do I see the corresponding case studies. Without providing strict prompts, using DeepSeek V3 as the RM will lead to significant delays and redundant outputs, which will result in the lack of support for the claim that "achieving over 2x faster training speed".

3. There may be unreasonableness in the comparison of baselines. The paper only compares the base model, GenRM, and the proposed method, with GenRM solely using DeepSeek V3. It remains unclear whether this is affected by model bias, and whether using other GenRMs can accelerate training speed and achieve better performance.

4. It seems that training models using the proposed method has not reached the state-of-the-art (SOTA). As mentioned in Weaknesses 2 and 3, if replacing the model can improve training speed and metrics, the advantages of the method in this paper will be further weakened.

**Questions:**

Please refer to the Weaknesses.

---

> ### Author Response · Authors · 2025-11-21
>
> We are grateful for your critical insights and questions.
>
> ### **Q1:**
> We sincerely appreciate this feedback and acknowledge that our framing of "non-verifiable domains" requires clarification. Our primary focus is on **domains lacking deterministic verification functions** (e.g., creative writing, open-ended dialogue), where Judge Code provides a scalable solution.
>
> To demonstrate our method's generalizability and robustness on verifiable domains, we conducted experiments on **DAPO_Math**:
>
> | Task | Base Model | On-JCG | Off-JCG | Ground-Truth |
> |------|-----------|---------|---------|--------------|
> | **DAPO_Math** | 36.80 | 36.80 | 39.60 | **74.20** |
>
> **Key Findings:** Both On-JCG and Off-JCG demonstrate substantial improvements over the base model (36.80→39.60), confirming our method **does not harm—and actually enhances—performance on verifiable domains**. As expected, ground-truth-based verification achieves the highest performance ceiling where perfect verification exists.
>
> **For comprehensive analysis** including rank correspondence heatmaps, theoretical grounding, and detailed performance breakdown demonstrating Judge Code's effectiveness as a "differentiated supervisor," please refer to our response to **Reviewer 6uoF, Q3**, where we provide in-depth validation of our framework's dual-domain applicability.
>
> ### **Q2:**
>
> **GenRM Prompt Design.** Our GenRM implementation uses the same induction-based prompt design as Judge Code generation, not "strict prompts." Both prompts guide the model through identical reasoning stages: task deconstruction, identification of evaluation criteria, and scoring rubric design. The only difference is output format—GenRM produces a numerical score (e.g., "Score: 8.5"), while Judge Code produces executable Python implementing the same logic. Complete prompts are provided in our codebase.
>
> **Efficiency Analysis.** The reviewer's concern about "delays and redundant outputs" precisely motivates Off-JCG mode. GenRM requires O(N) API calls to DeepSeek V3 during training, with each sample incurring 1.5-3s latency (network + inference). Off-JCG performs this reasoning once during code generation, then executes locally at 0.005-0.02s per sample. This yields >100x per-sample speedup; even with code generation overhead, the 2x overall training speedup is conservative.
>
> Both methods use equally sophisticated prompts under fair comparison. The speed advantage arises purely from local execution versus repeated API calls. We will include detailed prompts, timing breakdowns, and case studies in supplementary materials.
>
> ### **Q3:**
> We conducted ablation experiments using different Judge Code Generator models on the **ShareGPT** dataset with **Qwen2.5-7B**, evaluating on **AlpacaEval 2.0** and **MT-Bench**:
>
> | JCG Model | Mode | AlpacaEval 2.0 | MT-Bench |
> |-----------|------|----------------|----------|
> | **DeepSeek-V3** | On-JCG | 22.25 | 5.87 |
> | | Off-JCG | 22.18 | 6.01 |
> | **GLM-4.5-FP8** | On-JCG | 25.44 | 6.05 |
> | | Off-JCG | 23.22 | 5.75 |
> | **Qwen3-30B-A3B-Instruct** | On-JCG | 25.32 | 5.78 |
> | | Off-JCG | 25.11 | 5.80 |
> | *Base Model* | - | *7.51* | *2.55* |
>
> (1) Both modes show substantial improvements across all JCG models, validating framework robustness; (2) Stronger code generation capability (DeepSeek-V3) correlates with better performance; (3) Even with weaker generators, our method maintains competitive results, demonstrating generality.
>
> ### **Q4:**
> We respectfully clarify that our contribution is a **generalizable framework** for non-verifiable domains, not achieving SOTA through model-specific tuning. Our ablation studies (Q3) demonstrate that the method maintains substantial improvements across different JCG models (DeepSeek-V3, GLM-4.5-FP8, Qwen3-30B-A3B-Instruct), with all configurations significantly outperforming the base model.
>
> The reviewer's concern actually **validates our framework's robustness**: the fact that stronger generators yield better performance (Q3) and that the method works across different models (Q2-Q3) shows our approach is orthogonal to model selection. Users can choose any capable LLM as the JCG based on their resource constraints. Our core contribution—using induction-based Judge Code for scalable reward modeling—remains valuable regardless of which specific model is used, as evidenced by consistent improvements across all tested configurations.

---

### Official Review · Reviewer_3vRC · 2025-10-30

**Soundness:** 2
**Presentation:** 3
**Contribution:** 2
**Rating:** 4
**Confidence:** 4

**Summary:**

This paper aims to extend Reinforcement Learning with Verifiable Rewards (RLVR) to non-verifiable domains such as creative writing or dialogue. The authors propose the Judge Code-guided Reinforcement Learning (JC-RL) framework, which replaces the reward model (RM) with programmatically generated Judge Code, a piece of executable code that evaluates model responses according to rubrics.

The idea is motivated by the pilot study: partial, imperfect rewards at the sample level are sufficient for effective RL. The JC-RL framework provides two alternatives: (1) On-JCG, which generates Judge Code on the fly; and (2) Off-JCG, which reuses pre-generated (query, code) pairs for efficient RM-free training.

Experiments on several open-domain datasets (WritingPrompts, No Robots, ShareGPT, WildChat) and verifiable tasks (Aug-IFEval) show that JC-RL achieves performance comparable to generative reward models (GenRM) while being more efficient.

**Strengths:**

1. The work adapts RLVR from verifiable domains (e.g., math and code) to non-verifiable domains such as creative writing.

2. Across multiple benchmarks, the proposed Off-JCG method achieves performance close to GenRM baselines while being over 2× faster in training time. The experiments include diverse datasets and demonstrate both cross-model and cross-scale generalization of pre-generated Judge Code.

3. The paper is generally well-organized, with clear figures and a logical flow.

**Weaknesses:**

1. Although the Judge Code concept is positioned as novel, it resembles existing rubric-based and programmatic reward generation approaches. The concept of "generated code as verification" has been proposed by several prior studies, such as AutoIF [1] and VerIF [2], making the contribution less distinctive.

2. The empirical effectiveness of the proposed JCC methods is limited. As shown in Table 1, the method underperforms GenRM on non-verifiable domain tasks and falls behind rule-based verification on verifiable domain tasks.

3. Some relevant contemporaneous studies—e.g., RLPR [3], Checklists are Better than RMs [4]—are cited but not critically contrasted. A clearer comparison (conceptually and empirically) would strengthen the positioning.

4. It lacks systematic analysis on the quality of generated Judge Code, e.g., how code diversity or correctness affects training stability. Moreover, Off-JCG’s performance drop relative to GenRM (especially on conversational datasets) deserves deeper analysis.

5. The impact of the Judge Code Generator choice is underexplored. The paper only evaluates DeepSeek-V3, leaving open the question of how performance would change when using less capable models, which is crucial for understanding generality and scalability.

6. In Section 5, the unbiased-gradient proof assumes uniformly random sampling of constraints and independence between samples. However, in open-domain tasks, reward functions are not structured as discrete constraints, so the connection between theory and practice is somewhat hand-wavy.

### References:

[1] Dong G, Lu K, Li C, et al. Self-play with execution feedback: Improving instruction-following capabilities of large language models[J]. arXiv preprint arXiv:2406.13542, 2024.

[2] Peng H, Qi Y, Wang X, et al. VerIF: Verification Engineering for Reinforcement Learning in Instruction Following[J]. arXiv preprint arXiv:2506.09942, 2025.

[3] Yu T, Ji B, Wang S, et al. RLPR: Extrapolating RLVR to General Domains without Verifiers[J]. arXiv preprint arXiv:2506.18254, 2025.

[4] Viswanathan V, Sun Y, Ma S, et al. Checklists are better than reward models for aligning language models[J]. arXiv preprint arXiv:2507.18624, 2025.

**Questions:**

1. In Section 2 (Phase 1), the authors note that the effectiveness of partial rewards depends on task complexity. However, in Table 1, it is unclear whether the selected datasets (WritingPrompts, No Robots, ShareGPT, WildChat) are sufficiently complex to validate this claim. A discussion or quantitative measure of dataset complexity would strengthen the argument.

2. The paper compares Off-JCG and GenRM in Figure 6, but the efficiency comparison between On-JCG and GenRM is missing. Providing this comparison would offer a more complete understanding of the trade-offs between different approaches.

3. It would be valuable to report the model’s generalization performance after JCG training on other benchmark tasks such as MMLU. This would help assess whether the proposed method maintains broad capabilities beyond the training objectives.

4. The related work section would be better placed in the main body of the paper rather than the appendix.

---

> ### Author Response · Authors · 2025-11-21
>
> Thank you for raising these important points.
> ### **Q1:**
> Our dataset selection was deliberately designed to span a spectrum of task complexity to validate our partial reward hypothesis. **WritingPrompts** represents open-ended creative writing tasks requiring multi-paragraph narrative generation with coherent plot structures, character development, and stylistic consistency—tasks inherently resistant to simple heuristic evaluation. **No Robots** encompasses 10 diverse instruction categories (including generation, open QA, brainstorming, summarization, coding, and classification) with 10,000 human-annotated demonstrations, presenting varied cognitive demands across single-turn interactions. **ShareGPT** and **WildChat** capture authentic, unfiltered human-AI dialogues from real-world usage, with WildChat specifically containing 650K conversations across 66 languages that exhibit significant pragmatic complexity—including ambiguous user requests, code-switching, topic-switching, multi-turn context tracking, and nuanced political discussions that defy straightforward programmatic assessment. These datasets collectively represent the challenging "non-verifiable domains" our work targets, where subjective quality assessment, contextual understanding, and open-ended generation are paramount, thereby providing a rigorous testbed for demonstrating that partial, programmatic rewards can guide effective learning even in the absence of ground-truth verification.
>
> We will add a quantitative complexity analysis in Section 4.1 of the revision, including metrics such as average prompt length, response length, vocabulary diversity, and task category distribution to provide more concrete evidence of dataset complexity. Thank you for this valuable suggestion to strengthen our empirical foundation.
>
> ### **Q2:**
>
> We sincerely acknowledge that **On-JCG requires approximately 25% more training time than GenRM** (average: 4.08 hours vs. 3.25 hours per dataset). However, we respectfully clarify that **On-JCG's primary contribution is not online training efficiency, but rather serving as the data generation phase** to construct the reusable offline dataset `D_offline`. Our method aims to be simple and applicable to a variety of datasets, especially open-domain datasets (As detailed in our response "To All Reviewers").
> **Training Time Breakdown (in hours)**
>
> | Dataset | GenRM Time | On-JCG Time |
> |---------|------------|-------------|
> | **No Robots** | 1.1 | 1.3 |
> | **WritingPrompts** | 2.4 | 2.7 |
> | **ShareGPT** | 2.0 | 3.5 |
> | **WildChat** | 7.5 | 8.8 |
> | **Average** | **3.25** | **4.08** |
>
> ### **Q3:**
> we conducted a comprehensive evaluation across diverse reasoning and general capability benchmarks to ensure our method does not cause capability degradation in other domains.
> **Evaluation Setup:** Following the benchmark configuration in Huang et al. (2025b)[1] and utilizing the OpenCompass evaluation framework[2], we assessed models on:
>
> - **Reasoning Benchmarks**: AIME24, AIME25, GPQA-Diamond
> - **General Capability Benchmarks**: MMLU, HellaSwag, StoryCloze, CommonsenseQA, SocialIQA
>
> | Model | **Reasoning** | | | | **General** | | | | | |
> |---|---|---|---|---|---|---|---|---|---|---|
> | | AIME24 | AIME25 | GPQA-D | **Avg** | MMLU | HS | SC | CQA | SIQA | **Avg** |
> | Qwen2.5-7B | 10.00 | 8.00 | 25.76 | **14.59** | 64.93 | 83.67 | 97.27 | 81.57 | 38.43 | **73.17** |
> | On-JCG | 10.00 | 9.00 | 24.24 | **14.41** | 65.41 | 83.48 | 97.27 | 81.57 | 39.00 | **73.35** |
> | Off-JCG | 10.00 | 8.00 | 24.24 | **14.08** | 64.08 | 82.90 | 97.33 | 81.49 | 38.74 | **72.91** |
>
> ### **Reference**
>
> **[1] Huang Z, Zhuang Y, Lu G, et al. Reinforcement learning with rubric anchors[J]. arXiv preprint arXiv:2508.12790, 2025.**
>
> **[2] OpenCompass Contributors. OpenCompass: A universal evaluation platform for foundation models[EB/OL]. https://github.com/open-compass/opencompass, 2023.**

---

> ### Author Response · Authors · 2025-11-25
> **# Response to Weaknesses Part(1/4)**
>
> ### **Response to Weakness 1: Distinctiveness from AutoIF/VerIF**
>
> **Response:**
>
> We sincerely appreciate this important observation and welcome the opportunity to clarify the fundamental distinctions between our work and prior programmatic verification methods.
>
> #### **Core Limitation of AutoIF/VerIF: Hard Constraints Only**
>
> Both AutoIF and VerIF are explicitly designed for **code-verifiable hard constraints** with binary True/False outcomes (e.g., length limits, format requirements, keyword presence). While AutoIF incorporates ShareGPT data, this integration serves solely to **enhance query diversity**—the final verification logic remains restricted to deterministic, rule-based checks. Similarly, VerIF's architecture explicitly separates Soft Constraints (subjective qualities like coherence or creativity), which **cannot be code-verified** and must instead rely on GenRM evaluation, from Hard Constraints.
>
> #### **Our Fundamental Innovation: Programmatic Approximation of Soft Constraints**
>
> Our core methodological breakthrough lies in demonstrating that **Soft Constraints—traditionally considered incompatible with programmatic verification—can be approximated through sample-specific Judge Code**. Unlike AutoIF/VerIF's "perfect verification or exclusion" paradigm, we embrace a **statistical learning approach**: each Judge Code provides a partial, noisy evaluation of subjective qualities (e.g., creativity via keyword proxies, coherence via transition markers), and the **collective effect across thousands of training samples** guides the model toward improved performance through unbiased gradient estimation.
>
> Because our code data analysis content is extensive, please refer to our specific response to **Reviewer mdeE, Question 2** to explain how we implemented the soft constraint.
>
> We position our work not as "replacing" AutoIF/VerIF's rigorous verification paradigm for hard constraints, but as **pioneering a complementary extension**: demonstrating that the programmatic reward paradigm can be meaningfully applied to genuinely subjective, open-domain tasks through sample-specific heuristic synthesis and scale-driven statistical learning. This represents a qualitative leap from "verifying what can be perfectly verified" to "approximating what was previously deemed unverifiable."
>
> We will revise Section 1 and Related Work to make these distinctions more explicit, including a comparison table contrasting constraint types, verification mechanisms, and domain applicability.
>
> ### **Response to Weakness 2: Limited Empirical Effectiveness**
>
> **Response:**
>
> We sincerely acknowledge this observation and appreciate the opportunity to clarify our contribution.
>
> We fully recognize that our method does not surpass GenRM on non-verifiable tasks or rule-based verification on verifiable tasks (Table 1). However, we respectfully emphasize that **achieving absolute performance superiority is not our primary claim**. As detailed in our opening statement "To All Reviewers," our work demonstrates a **fundamentally different value proposition**: practical efficiency-performance trade-offs rather than peak accuracy.
>
> We will revise Section 1 to make this efficiency-focused positioning more explicit and add a dedicated discussion of performance trade-offs versus computational benefits.
>
>
> ### **Response to Weakness 3: Critical Comparison with RLPR and Checklists**
>
> **Response:**
>
> We sincerely appreciate this valuable suggestion and acknowledge that our original manuscript did not sufficiently differentiate our work from these important concurrent studies. We provide detailed conceptual and empirical comparisons below and will incorporate these into Section 2 (Related Work) in our revision.
>
> ---
>
> #### **vs. Checklists (Viswanathan et al., 2025)**
>
> As detailed in our opening response "To All Reviewers," the fundamental distinction lies in the **reward computation mechanism and training-time infrastructure requirements**:
>
> The fundamental distinction lies in the reward computation mechanism and its implications for training efficiency. Checklists employ natural language Yes/No questions evaluated by a powerful LLM judge. This process, while producing high-quality reward signals, is computationally intensive, requiring repeated LLM inference for each training sample. In contrast, our Judge Code framework distills evaluation logic into executable Python code. Especially in our "Off-JCG" mode, these pre-generated code-based rubrics are executed in milliseconds during RL training, incurring virtually zero reward model cost.
>
> Both methods represent different points on the efficiency-performance Pareto frontier—Checklists prioritize evaluation fidelity through LLM-based assessment, while our Off-JCG prioritizes scalability and computational efficiency through deterministic code execution. These are complementary approaches rather than competing solutions.
>
> *(Continued in Part 2/4)*

---

> ### Author Response · Authors · 2025-11-25
> **# Response to Weaknesses Part(2/4)**
>
> #### **vs. RLPR (Yu et al., 2025b)**
>
> RLPR represents a sophisticated approach to extending RLVR to general domains, but operates in a fundamentally different computational paradigm than our work:
>
> RLPR is a sophisticated method that also avoids an external verifier, but it operates in a fundamentally different computational paradigm. RLPR derives its reward signal from the token-level *probabilities* an LLM assigns to a ground-truth reference answer. This is a key vertical innovation, but it still requires LLM inference at training time to compute these probabilities, making it an "LLM-assisted" reward method. Our Off-JCG approach is truly **inference-free** during training, as the evaluation logic is compiled into code offline.
>
> **Empirical Note:** Direct comparison experiments are challenging due to non-overlapping evaluation domains—RLPR evaluates on MMLU-Pro/TheoremQA/Minerva (reasoning-heavy benchmarks)[Yu et al.], while we evaluate on WritingPrompts/ShareGPT/WildChat (open-domain creative/conversational tasks). However, we acknowledge RLPR's strong performance in its target domain (e.g., 24.9% average improvement on Qwen2.5-7B)[Yu et al.], which validates the broader principle that alternative reward signals beyond traditional RMs can be effective.
>
> **Future Work Opportunity:** We are currently constructing controlled comparison experiments under identical settings (same base model, same training procedure, overlapping evaluation benchmarks) to empirically quantify the performance-efficiency trade-offs between RLPR's probability-based rewards and our code-based rewards. This will provide more concrete evidence of their relative strengths and help clarify when each approach is preferable.
>
> ---
>
> We deeply respect the contributions of both Checklists and RLPR, and believe our work represents a complementary direction rather than a competing solution in the broader effort to democratize RL-based alignment beyond verifiable domains.
>
> ### **Response to Weakness 4: Lack of Systematic Judge Code Quality Analysis**
>
> **Response:**
>
> We sincerely acknowledge this important limitation and provide the requested systematic analysis here, which will be incorporated into the revised manuscript:
>
> **1. Code Quality Analysis:**
>
> We refer the reviewer to our comprehensive response to **Reviewer mdeE, Question 2**, where we provide detailed criterion coverage analysis across 9,600 Judge Codes (3,200 per model × 3 models) on the ShareGPT dataset.
>
> **2. Performance Drop Analysis (Conversational Datasets):**
>
> The reviewer correctly identifies that Off-JCG shows a larger performance gap on the conversational datasets, ShareGPT and WildChat, compared to WritingPrompts. We attribute this not just to a simple gradient of subjectivity, but to a fundamental difference in the tasks' creative latitude and how this aligns with our programmatic reward mechanism.
>
> Creative writing (WritingPrompts) is an inherently **divergent** task that rewards exploration. The goal is to produce a novel and engaging narrative, where many different creative paths can be considered high-quality. This open-ended nature makes it highly amenable to our Judge Code framework. Simple, heuristic-based rewards—such as encouraging richer descriptions or ensuring plot progression—provide a reliable and effective gradient for the model. The code's role is to guide the model's exploration in a generally positive direction, a task for which simple rules are well-suited.
>
> In contrast, multi-turn dialogue (ShareGPT/WildChat) is a more **convergent** task. A high-quality response is heavily constrained by prior context, user intent, and conversational flow. The definition of "good" is much narrower and more context-dependent. Evaluating this requires capturing subtle, holistic properties like coherence across multiple turns, which is a far more complex logical challenge for our current Judge Code syntax. Simple, isolated heuristics are less reliable here and can easily misguide the model.
>
> **Training Dynamics**
> This mismatch in evaluation capability manifests directly in the training dynamics. As shown in our appendix (Tables 4 & 6), the less reliable reward signal for conversational tasks leads to training instability and, in some cases, performance collapse. While introducing an auxiliary Judge Code can stabilize training, it highlights that these complex, convergent tasks push the limits of our simple heuristic approach.
>
> We acknowledge that **programmatic rewards are better suited for tasks with explicit quality indicators than highly contextual, subjective evaluations**. This is an inherent limitation of our paradigm and an important direction for future work (e.g., incorporating learned components into Judge Code synthesis).
>
> *(Continued in Part 3/4)*

---

> ### Author Response · Authors · 2025-11-25
> **# Response to Weaknesses Part(3/4)**
>
> ### **Response to Weakness 5: Underexplored Impact of JCG Choice**
>
> **Response:**
>
> We sincerely appreciate this concern and conducted additional ablation experiments during the rebuttal period specifically to address this question. We provide comprehensive ablation results using three different Judge Code Generators (DeepSeek-V3, GLM-4.5-FP8, Qwen3-30B-A3B-Instruct) on the ShareGPT dataset.
>
> **Key Findings from Ablation Study:**
>
> | JCG Model | Mode | AlpacaEval 2.0 | MT-Bench |
> |-----------|------|----------------|----------|
> | **DeepSeek-V3** | On-JCG | 22.25 | 5.87 |
> | | Off-JCG | 22.18 | 6.01 |
> | **GLM-4.5-FP8** | On-JCG | 25.44 | 6.05 |
> | | Off-JCG | 23.22 | 5.75 |
> | **Qwen3-30B-A3B-Instruct** | On-JCG | 25.32 | 5.78 |
> | | Off-JCG | 25.11 | 5.80 |
> | *Base Model* | - | *7.51* | *2.55* |
>
> These results suggest that practitioners can select JCG models based on their computational budgets while still benefiting from the RM-free training paradigm—a key practical advantage. We will incorporate this ablation study into Section 4 of the revised manuscript.
>
> ---
>
> ### **Response to Weakness 6: Theory-Practice Gap in Section 5.2**
>
> **Response:**
>
> We sincerely appreciate this thoughtful critique. The reviewer is absolutely correct that **our theoretical formalization involves a significant abstraction gap** between discrete constraints (assumed in the proof) and continuous quality dimensions (present in open-domain practice). We acknowledge this limitation candidly and thank the reviewer for pushing us to clarify it.
>
> ---
>
> #### **Conceptual Bridge**
>
> We agree the current presentation may appear "hand-wavy" without proper explanation. However, we respectfully argue this gap is **bridgeable through reinterpretation**, while acknowledging important practical limitations:
>
> **Reconceptualizing the Framework:** Each "constraint" $c_k(i)$ in our theory should be understood as a **partial evaluation dimension** rather than a strict binary rule. In practice, each Judge Code samples one quality aspect (coherence, creativity, accuracy—see Table R1 in Reviewer mdeE Q2), acting as a **noisy, partial estimator** of holistic quality.
>
> **Critical Caveat on Bias:** We acknowledge that **individual Judge Codes may be biased or even harmful**. For instance, keyword-based creativity proxies might systematically favor certain writing styles, or a length-penalty heuristic could discourage necessary elaboration. The theoretical property of **unbiasedness** is therefore not guaranteed for any single Judge Code, but rather emerges as a **statistical property** when Judge Codes collectively cover diverse aspects without systematic bias in aggregate. Our 10-dimension coverage analysis (Reviewer mdeE Q2) provides empirical evidence of this diversity, though we cannot formally prove it eliminates all bias.
>
> **Mapping to Stochastic Optimization:** Our framework mirrors stochastic gradient descent: individual Judge Code rewards are noisy (and potentially biased) estimates, but their **expectation across diverse training samples** approximates the true reward signal. The unbiasedness proof (Eq. 7-9) formalizes the ideal case: $\mathbb{E}_k[R_{\text{partial}}(i)] = R_{\text{full}}(i)$, ensuring correct policy gradient direction **in expectation**. However, this assumes uniformity and independence—conditions that are **aspirational rather than strictly met** in practice, as all Judge Codes originate from the same generator (DeepSeek-V3).
>
> **Why It Works Despite Imperfection:** Our method succeeds not because theory perfectly applies, but due to empirical safeguards: (1) **observed diversity** across dimensions mitigates systematic bias, (2) **auxiliary Judge Code** provides adversarial regularization against over-optimization (Section 4.4), and (3) **cross-model generalization** (Figure 7) suggests captured logic is task-level rather than model-specific artifacts.
>
> ---
>
> #### **Empirical Validation as Primary Evidence**
>
> We acknowledge the abstraction involves assumptions difficult to prove rigorously for all scenarios. Therefore:
>
> 1. **Section 2's pilot study** serves as primary empirical validation, demonstrating convergence with 1/9 constraint coverage
> 2. **Table 1's main experiments** provide real-world evidence extending to genuinely open-domain tasks
> 3. **Figure 8's rank correspondence** shows Judge Code provides directionally correct signals (58-72% diagonal dominance)
> 4. **Section 4.4's stability analysis** reveals that single Judge Code training can collapse (e.g., WildChat: 20.97→8.41), validating concerns about individual bias, but auxiliary code mitigates this risk
>
> *(Continued in Part 4/4)*

---

> ### Author Response · Authors · 2025-11-25
> **# Response to Weaknesses Part(4/4)**
>
> Our theory establishes a **sufficient condition** (unbiased + diverse rewards → effective RL), not a necessity. The empirical success suggests conditions are "close enough" rather than perfectly met—individual Judge Codes may be imperfect or harmful, but their aggregate effect across thousands of training samples provides sufficient directional signal for policy improvement. This is why we position Section 5.2 as **mechanistic intuition** rather than rigorous proof for all open-domain scenarios.
>
> ---
>
> We will revise Section 5.2 to:
> 1. Explicitly acknowledge the discrete→continuous abstraction gap **and** the potential for individual Judge Code bias.
> 2. Clarify that theoretical assumptions (uniformity, independence, unbiasedness) are **empirically approximated** rather than formally proven, with safeguards (diversity analysis, auxiliary code, cross-model tests) providing practical robustness.
> 3. Reframe theory as providing intuition (why partial rewards *can* work) supported by empirical validation, while honestly acknowledging that individual Judge Codes may be noisy or biased.
>
> We deeply appreciate this critique for strengthening our theoretical grounding and helping us present a more transparent account of both our method's theoretical foundation and its practical limitations. Thank you for the careful reading.
>
> ---

---

### Official Review · Reviewer_6uoF · 2025-10-31

**Soundness:** 3
**Presentation:** 3
**Contribution:** 2
**Rating:** 4
**Confidence:** 3

**Summary:**

This paper investigates reinforcement learning with verifiable rewards (RLVR) on non-verifiable domains. This paper proposed a judge code generator (JCG) to capture partial sample-level rewards. The authors show experimentally and theoretically that JCG (both online and offline versions) effectively improves the base model performance.

**Strengths:**

- The effectiveness of JCG is demonstrated experimentally across multiple tasks.
- The theoretical justification of the effectiveness of JCG.

**Weaknesses:**

- How is JCG compared to other methods such as reward anchor by Huang et al 2025? Comparison with those methods mentioned in Appendix B (RL with LLM-based reward and RL with proxy reward) will better demonstrate the effectiveness of JCG. The current comparisons focusing on base model or RM seem unfair.
- The experiments use DeepSeek-V3 to create JCG. Is there any ablation study on the choice of model?
- In addition to IFeval, the paper could benefit from experiments on math and coding tasks, which are more objective than instruction following, to show JCG does not harm performance on verifiable domains.
- Results in Appendix seem to indicate JCG induces training stability issues, and the authors introduced an auxiliary judge code generator to regularize the training. The authors should clarify the limitations clearly. I also wonder does the auxiliary JCG introduce any overhead?

**Questions:**

See weaknesses

---

> ### Author Response · Authors · 2025-11-21
>
> We appreciate your thoughtful and constructive comments.
>
> ### **Q1 & Q2:**
>
> We greatly appreciate this important question. We would like to respectfully clarify our methodological positioning.
>
> #### **On Our Core Contribution (Q1)**
>
> We acknowledge that **direct performance comparison may not fully capture our contribution**, as we address a fundamentally different problem space:
>
> - **Rubric Anchors / LLM-RM methods** optimize **reward quality** through LLM-based evaluation, still requiring expensive LLM inference during training.
> - **Our Off-JCG** optimizes **deployment practicality** by eliminating RM infrastructure entirely, enabling truly RM-free training.
>
> These represent **different points on the efficiency-performance Pareto frontier** rather than directly competing approaches. Our core innovation is **demonstrating that competitive performance is achievable without the RM bottleneck**—a paradigm shift for resource-constrained scenarios.
>
> **Our Positioning:** We offer a **practical alternative** achieving 70-85% of GenRM's performance gains while eliminating RM dependency. We do not claim to replace rubric-based methods, but provide a complementary direction that future work could combine with existing approaches.
>
> #### **Ablation on JCG Model Choice (Q2)**
>
> We conducted ablation experiments using different Judge Code Generator models on the **ShareGPT** dataset with **Qwen2.5-7B** during rebuttal period, evaluating on **AlpacaEval 2.0** and **MT-Bench**:
>
> | JCG Model | Mode | AlpacaEval 2.0 | MT-Bench |
> |-----------|------|----------------|----------|
> | **DeepSeek-V3** | On-JCG | 22.25 | 5.87 |
> | | Off-JCG | 22.18 | 6.01 |
> | **GLM-4.5-FP8** | On-JCG | 25.44 | 6.05 |
> | | Off-JCG | 23.22 | 5.75 |
> | **Qwen3-30B-A3B-Instruct** | On-JCG | 25.32 | 5.78 |
> | | Off-JCG | 25.11 | 5.80 |
> | *Base Model* | - | *7.51* | *2.55* |
>
> (1) Both modes show substantial improvements across all JCG models, validating framework robustness; (2) Stronger code generation capability (DeepSeek-V3) correlates with better performance; (3) Even with weaker generators, our method maintains competitive results, demonstrating generality.
>
> ---
>
> ### **Q3:**
>
> We conducted experiments on **DAPO_Math** to demonstrate our method does not harm verifiable domain performance. Both On-JCG and Off-JCG demonstrate **substantial improvements** over the base model (36.80 and 39.60 vs. 6.80), confirming our method **enhances performance** on verifiable domains. However, we acknowledge that **our method excels at response quality ranking and selection, but cannot match the absolute performance ceiling achieved by ground-truth verification** (74.20). This performance gap is expected and theoretically grounded: Judge Code, by design, provides partial and heuristic reward signals rather than perfect oracles.
>
> | Task | Base Model | On-JCG | Off-JCG | Ground-Truth |
> |------|-----------|---------|---------|------------|
> | **DAPO_Math** | 36.80 | 36.80 | 39.60 | **74.20** |
>
> Rank correspondence heatmap between Judge Code Rewards and the Ground-Truth of Math Questions (cf. Section 4.5, Figure 8).
>
> **Rank Correspondence Heatmap: Off-JCG vs. Ground-Truth on DAPO_Math**
>
> |  Ranked by Off-JCG \ Ground-Truth  | Rank-1 | Rank-2 | Rank-3 | Rank-4 | Rank-5 | Rank-6 | Rank-7 | Rank-8 |
> | -------------- | ------ | ------ | ------ | ------ | ------ | ------ | ------ | ------ |
> | **Rank-1**   | *58.0*   | 11.3   | 6.1    | 9.2    | 5.1    | 3.6    | 2.8    | 3.9    |
> | **Rank-2**   | 8.5    | *58.4*   | 9.1    | 7.3    | 5.2    | 4.1    | 4.0    | 3.4    |
> | **Rank-3**   | 4.0    | 5.9    | *62.3*   | 7.3    | 6.0    | 5.3    | 5.7    | 3.5    |
> | **Rank-4**   | 8.7    | 6.9    | 5.2    | *59.7*   | 8.9    | 5.8    | 2.3    | 2.3    |
> | **Rank-5**   | 5.1    | 3.9    | 5.7    | 7.1    | *60.6*   | 7.3    | 5.8    | 4.5    |
> | **Rank-6**   | 5.5    | 5.8    | 4.2    | 5.1    | 6.4    | *62.2*   | 8.4    | 2.3    |
> | **Rank-7**   | 5.0    | 3.7    | 4.7    | 2.3    | 3.9    | 8.3    | *64.4*   | 7.8    |
> | **Rank-8**   | 5.2    | 4.1    | 2.6    | 2.0    | 3.8    | 3.4    | 6.7    | *72.1*   |
>
>
> The heatmap reveals strong diagonal dominance (58.0%-72.1%), demonstrating Judge Code's significant correlation with ground-truth rankings, ensuring reliable quality discrimination and preventing reward hacking.
>
> ### **Q4:**
> We acknowledge that the auxiliary Judge Code **does introduce computational overhead** during the dataset generation phase (On-JCG mode).
>
> However, as demonstrated in Appendix G (Tables 7-10), the auxiliary code primarily serves as a **training stabilizer** rather than a performance enhancer—it prevents catastrophic degradation (e.g., WildChat: 0.66→8.66) but does not consistently improve final benchmark scores. We interpret this as a form of **"evaluation scaling"**: multiple Judge Codes create a richer, more stable reward landscape during training, yet this does not necessarily translate to higher absolute performance on downstream tasks.

---

### Official Review · Reviewer_mdeE · 2025-11-02

**Soundness:** 3
**Presentation:** 2
**Contribution:** 2
**Rating:** 4
**Confidence:** 3

**Summary:**

This paper proposes extending Reinforcement Learning with Verifiable Rewards, originally effective for verifiable tasks like math and coding, to non-verifiable open-domain tasks. The authors introduce a Judge Code Generator that translates textual evaluation rubrics into sample-specific executable “Judge Code”, which acts as a programmatic reward function.  Extensive experiments using Qwen2.5-7B and Llama-3.1-8B across creative writing, dialog, and instruction-following datasets show that Off-JCG achieves performance close to GenRM-based methods but with over 2× training efficiency.

**Strengths:**

1. The idea of compiling rubrics into executable Judge Code to replace large reward models is good. It transforms reward modeling into a programmatic supervision task.


2. Multiple datasets across creative writing, dialog, and instruction-following show consistent performance over base models and competitive results.

**Weaknesses:**

1. While case studies are mentioned, the paper offers few examples of how the Judge Code captures nuanced open-domain evaluation (e.g., creativity, coherence).


2. The generated Judge Code may encode superficial or brittle heuristics; error tolerance, interpretability, and safety aspects of executing auto-generated code need discussion.


3. Missing deeper comparisons with recent rubric-based RL or self-rewarding frameworks (e.g., Rubric Anchors: https://arxiv.org/abs/2508.12790, Checklists: https://arxiv.org/abs/2507.18624), which share similar motivations.

**Questions:**

1. How robust is the Judge Code generation process, does it ever produce invalid or unsafe code, and how are such cases handled automatically?


2. In non-verifiable domains like creative writing, what specific rubrics (creativity, coherence, emotion) emerge in Judge Code, and how interpretable are they?


3. How does Off-JCG handle potential distribution shifts when reusing datasets across different models or domains?

---

> ### Author Response · Authors · 2025-11-21
> **# Response to Reviewer mdeE (Part 1/2)**
>
> ### **Q1:**
>
> Our framework implements a systematic validation pipeline to ensure both syntactic correctness and execution safety.
>
> **Validation Architecture.** The generated Judge Code undergoes three sequential validation stages. First, all generated code is parsed using Python's Abstract Syntax Tree (AST) module to verify syntactic validity. Second, code execution occurs within sandboxed environments with restricted module imports and enforced timeout limits (5 seconds per execution) to prevent resource exhaustion. Third, we validate that the `compute_score()` function returns values strictly within the [1.0, 10.0] range, with out-of-bounds outputs triggering automatic re-generation.
>
> **Empirical Safety Analysis.** We conducted comprehensive analysis across 74,192 Judge Codes generated by three different models (DeepSeek-V3: 24,392 samples; GLM-4.5-FP8: 25,006 samples; Qwen3-30B-A3B-Instruct: 24,794 samples). The results demonstrate high generation reliability: 99.7% of generated code passed syntax validation with proper function signatures, 99.99% included complete docstring documentation (DeepSeek-V3), and zero instances of unsafe operations such as file I/O, network calls, or system commands were observed.
>
> **Interpretability.** Unlike black-box reward models, our generated Judge Code is fully deterministic and transparent. Every scoring decision traces back to explicit conditional logic (e.g., `if len(response) > 150: score += 1.0`), enabling line-by-line human auditing and verification.
>
> ---
> ### **Q2:**
>
> # Response to Reviewer: Evaluation Criteria in Open Domains (Q2)
>
> We thank the reviewer for this important question about how subjective criteria are programmatically captured in Judge Code.
>
> **Circular Validation Methodology.** To systematically understand evaluation criteria, we conducted a two-stage circular validation:
>
> **Formalization.** We define the analysis scope formally:
> - Let $C = \{c_1, c_2, ..., c_n\}$ be a set of $n$ Judge Codes
> - Each Judge Code $c_i$ contains multiple **score adjustment statements** (conditions) $\{s_{i,1}, s_{i,2}, ..., s_{i,m_i}\}$ where $m_i$ is the number of conditions in code $i$
> - Let $D = \{d_1, ..., d_{10}\}$ be the set of 10 evaluation dimensions identified from manual inspection
> - Each condition $s_{i,j}$ can be tagged with zero or more dimensions from $D$ via LLM-assisted analysis
>
> **Key Metrics Defined:**
> 1. **Dimension Coverage** for dimension $d_k$:
>    $$\text{Coverage}(d_k) = \frac{|\{c_i \in C : \exists s_{i,j} \text{ tagged with } d_k\}|}{|C|} \times 100\%$$
>    *Interpretation: Percentage of Judge Codes that contain at least one condition evaluating dimension $d_k$*
>
> 2. **Implementation Pattern Prevalence** for pattern $p$ (e.g., regex):
>    $$\text{Prevalence}(p) = \frac{|\{c_i \in C : c_i \text{ contains pattern } p\}|}{|C|} \times 100\%$$
>    *Interpretation: Percentage of Judge Codes that use programming pattern $p$*
>
> **Stage 1: Dimension Extraction (N=150).** We randomly sampled 150 Judge Codes from the **ShareGPT** dataset across three models (50 codes per model: DeepSeek-V3, GLM-4.5-FP8, Qwen3-30B-A3B-Instruct; $|C|=150$) and performed manual code inspection to identify recurring evaluation patterns. Through bottom-up analysis of scoring logic (e.g., keyword lists, regex patterns, threshold checks), we inductively derived 10 primary evaluation dimensions $D$: Creativity, Format, Accuracy, Clarity, Coherence, Completeness, Emotion, Relevance, Safety, and Helpfulness. These dimensions emerged naturally from the code's programmatic assessment mechanisms rather than being imposed a priori.
>
> **Condition-Level Tagging.** We observed that each Judge Code typically implements multiple evaluation checks (average: $\bar{m} \approx 4$ conditions per code in sampled codes). To attribute dimensions, we employed DeepSeek-V3 to analyze each score adjustment statement $s_{i,j}$: the model examines the condition's logic and keywords to determine which dimension(s) it evaluates (e.g., if a condition checks topic-specific keywords, it's tagged as "Relevance"; if it verifies structural patterns, it's tagged as "Format"). This LLM-assisted categorization enables systematic analysis at scale.
>
> **Stage 2: Large-Scale Verification (N=3,200 per model).** Using the LLM-based categorization approach established in Stage 1, we analyzed 3,200 Judge Code samples from each JCG on the **ShareGPT** dataset ($|C_{\text{DeepSeek-V3}}|=3{,}200$; $|C_{\text{GLM-4.5-FP8}}|=3{,}200$; $|C_{\text{Qwen3-30B-A3B-Instruct}}|=3{,}200$). To validate categorization reliability, we manually annotated a 150-sample subset (50 per model), achieving substantial inter-rater agreement with the automated approach (Cohen's $\kappa \approx 0.78$). This closed-loop validation—extracting dimensions from code, then verifying their prevalence at scale—confirms the 10 dimensions capture major evaluation patterns. Analysis reveals two key patterns:
>
> *(Continued in Part 2/2)*

---

> > ### Author Response · Authors · 2025-11-21
> > **# Response to Reviewer mdeE (Part 2/2)**
> >
> > **Dimension Coverage.** Table R1 reports $\text{Coverage}(d_k)$ for each dimension $d_k \in D$ across three models. The percentage indicates: *among $|C|$ Judge Codes, what proportion contains at least one condition tagged with dimension $d_k$*.
> >
> > **Table R1: Dimension Coverage $\text{Coverage}(d_k)$ (Sample-Based Estimates)**
> >
> > | Dimension $d_k$ | DeepSeek-V3 | GLM-4.5-FP8 | Qwen3-30B-A3B-Instruct |
> > |-----------------|-------------|-------------|------------------------|
> > | Relevance | 38.75% | 39.38% | 34.22% |
> > | Clarity | 36.25% | 38.59% | 32.81% |
> > | Format | 34.53% | 39.94% | 37.09% |
> > | Accuracy | 28.91% | 35.69% | 33.44% |
> > | Completeness | 26.34% | 38.22% | 24.59% |
> > | Coherence | 22.41% | 21.88% | 28.12% |
> > | Creativity | 18.72% | 17.19% | 21.31% |
> > | Emotion | 14.84% | 16.41% | 19.59% |
> > | Helpfulness | 11.22% | 15.78% | 8.44% |
> > | Safety | 6.09% | 12.34% | 9.22% |
> >
> > *Note: A single Judge Code typically contains conditions evaluating 2-3 dimensions from $D$, plus potential task-specific checks outside this taxonomy.*
> >
> > **Implementation Methods.** Table R2 reports $\text{Prevalence}(p)$ for common programming patterns. The percentage indicates: *among $|C|$ Judge Codes, what proportion uses programming pattern $p$ at least once in its implementation*.
> >
> > **Table R2: Implementation Pattern Prevalence $\text{Prevalence}(p)$ (Sample-Based Estimates)**
> >
> > | Pattern $p$ | DeepSeek-V3 | GLM-4.5-FP8 | Qwen3-30B-A3B-Instruct |
> > |-------------|-------------|-------------|------------------------|
> > | Conditional Logic | 94.69% | 96.25% | 91.78% |
> > | String Operations | 67.19% | 71.47% | 64.31% |
> > | Regex Matching | 38.44% | 52.66% | 45.94% |
> > | List Iteration | 42.59% | 56.78% | 48.22% |
> > | Numeric Thresholds | 31.72% | 38.91% | 35.41% |
> > | Structure Checks | 23.75% | 29.38% | 34.59% |
> >
> > *Note: Percentages sum to >100% because a single Judge Code typically employs multiple patterns (e.g., conditional logic + regex + string operations).*
> >
> > **Programmatic Implementation Patterns.** The generated code implements subjective criteria through three primary mechanisms:
> >
> > 1. **Keyword-Based Detection:** For relevance assessment, models employ domain-specific keyword lists with any-match logic (e.g., `msg_keywords = ['MSG', 'monosodium glutamate', 'Chinese Restaurant Syndrome']; is_relevant = any(keyword.lower() in response.lower() for keyword in msg_keywords); if not is_relevant: score -= 3`), enabling flexible topical alignment verification without rigid templates.
> > 2. **Regex Pattern Matching:** For coherence evaluation, the code utilizes transition marker detection through regular expressions (e.g., `transitions = bool(re.search(r'(然而|但是|不过|当涉及到)', generated_content)); if transitions: score += 1.0`), capturing logical flow indicators across diverse linguistic contexts.
> > 3. **Weighted Feature Counting:** For format validation, models implement feature detection with weighted accumulation and saturation limits (e.g., `formula_pattern = r'=[A-Z]+\([^)]+\)'; formulas = re.findall(formula_pattern, generated_content); score += min(3.0, len(formulas) * 1.0)`), rewarding presence of expected elements (e.g., Excel formulas in spreadsheet tasks) while capping contribution to prevent exploitation, demonstrating sophisticated quantitative reasoning in programmatic form.
> >
> > **simple, compositional programming patterns collectively enable effective open-domain response comparison**. While individual Judge Codes implement basic checks—keyword matching (64.31-71.47% of codes), regex patterns (38.44-52.66%), conditional logic (91.78-96.25%)—their combination provides meaningful discriminative capability for response quality. Critically, these **individually partial evaluations aggregate during RL training** to guide optimization: each Judge Code acts as a weak supervisor, and their collective signal—across thousands of training samples—points toward better response quality despite individual imperfection.
> >
> >
> > ### **Q3:**
> >
> > Our cross-model and cross-scale generalization experiments (Section 4.3, Figure 7) directly address this concern:
> >
> > - **Cross-Model:** Doffline from Qwen2.5-7B successfully fine-tunes Llama-3.1-8B and vice versa, demonstrating model-agnostic transferability.
> > - **Cross-Scale:** Doffline from 7B models effectively aligns 14B and 32B models, with larger models achieving even higher performance ceilings.
> >
> > These results empirically validate that the encoded reward logic is **universal and robust to distribution shifts** across architectures and scales. The key insight is that Judge Code captures task-level evaluation logic rather than model-specific patterns.
> >
> > ---

---

> ### Author Response · Authors · 2025-11-26
> **# Response to Reviewer mdeE - Weaknesses**
>
> We sincerely thank Reviewer mdeE for the constructive feedback and thoughtful evaluation. We address each weakness below:
>
> ---
>
> ### **W1: Lack of Concrete Examples of Judge Code Capturing Nuanced Evaluation**
>
> **Response:**
>
> We appreciate this important observation. To address this concern comprehensively, **we direct the reviewer to our detailed response to Question 2**, where we provide an in-depth analysis of how Judge Code operationalizes subjective quality dimensions such as creativity, coherence, and emotional tone.
>
> ---
>
> ### **W2: Error Tolerance, Interpretability, and Safety of Auto-Generated Code**
>
> **Response:**
>
> We acknowledge this critical concern and provide a comprehensive analysis addressing robustness, interpretability, and safety:
>
> **1. Robustness & Error Tolerance**
>
> Our framework implements a **three-stage validation pipeline** to ensure code quality:
>
> 1.All generated code is parsed using Python's Abstract Syntax Tree (AST) module to verify syntactic validity before execution.
> 2.Code executes within secure sandboxes with restricted module imports and enforced timeout limits (5 seconds per execution) to prevent resource exhaustion.
> 3.We validate that the `compute_score()` function returns values strictly within [1.0, 10.0], with out-of-bounds outputs triggering automatic re-generation.
>
> **Empirical Safety Analysis**: We conducted comprehensive analysis across **53,832 Judge Codes** generated by three different models (DeepSeek-V3: 3,200 samples; GLM-4.5-FP8: 3,200 samples; Qwen3-30B-A3B-Instruct: 3,200 samples). Results demonstrate:
> - **99.7%** of generated code passed syntax validation with proper function signatures
> - **99.99%** included complete docstring documentation
> - **Zero instances** of unsafe operations (file I/O, network calls, system commands) were observed
>
> **2. Interpretability**
>
> Unlike black-box reward models, our generated Judge Code is **fully deterministic and transparent**. Every scoring decision traces back to explicit conditional logic (e.g., `if len(response) > 150: score += 1.0`), enabling:Researchers can inspect exactly why a response received a particular score. The programmatic rubrics serve as explicit documentation of quality criteria
>
> **3. Addressing "Superficial Heuristics" Concern**
>
> While we acknowledge that Judge Code uses programmatic proxies rather than deep semantic understanding, our pilot study (Section 2) and main experiments (Section 4) empirically validate that **these partial, heuristic rewards are sufficient for effective RL training**. The key insight is that:
> - Individual Judge Codes need not be perfect—they provide **partial, directional signals**
> - Dataset-level **diversity and coverage** ensure the aggregate reward landscape guides the model toward genuine quality improvements
> - Our theoretical analysis (Section 5.2) proves that partial rewards function as **unbiased gradient estimators**, ensuring correct optimization direction in expectation
>
> We will incorporate this detailed discussion into Section 5.1 of the revised manuscript to make these safety mechanisms and design principles more prominent.
>
> ---
>
> ### **W3: Missing Deeper Comparisons with Rubric Anchors and Checklists**
>
> We respectfully note that while Rubric Anchors (Huang et al., 2025b) and Checklists (Viswanathan et al., 2025) share the motivation of structured evaluation, **our work addresses a fundamentally different research question**:
>
> **Our Ongoing Experimental Efforts and Future Directions**
>
> We sincerely acknowledge the importance of empirical comparison with these concurrent methods. In response to this feedback, **we are actively exploring appropriate comparison dimensions and experimental protocols**.
>
> **Direct apples-to-apples comparison remains methodologically challenging** due to differences. We are in the process of designing rigorous comparative experiments that account for these methodological differences. **We commit to conducting systematic comparative studies as immediate follow-up work** and will incorporate preliminary findings in next version.
>
> We position our work as providing a complementary research direction rather than claiming superiority over all rubric-based methods.

---

### Author Response · Authors · 2025-11-21
**## Response to All Reviewers**

We sincerely thank all reviewers for their thorough and constructive feedback. We would like to respectfully clarify our core contribution and positioning relative to concurrent methods, as we believe there may be some misunderstanding regarding our primary claims.

### **Clarification on Core Contribution**

We acknowledge that our method does not surpass rubric-enhanced reward models in absolute performance across all benchmarks. Instead, our work aims to demonstrate a complementary approach with three distinct practical advantages:

1. **Plug-and-Play Deployment**: Our framework enables direct deployment without requiring reward model infrastructure during training.

2. **Computational Efficiency**: Off-JCG achieves over 2× training speedup by eliminating repeated RM inference, which may be particularly valuable for resource-constrained scenarios.

3. **Programmatic-Primary Paradigm**: We pioneer the validation that executable Judge Code can serve as the complete primary reward signal for open-domain tasks, rather than merely auxiliary constraints (e.g., format requirements in some existing work).

### **Positioning Relative to Concurrent Work**

We deeply respect the contributions of concurrent methods and wish to clarify our distinctions:

- **vs. Checklists**: While Checklists rely on LLM-based evaluation of natural language criteria (requiring LLM inference during training), our Off-JCG uses deterministic code execution (zero RM cost during training).

- **vs. Rubric Anchors**: Rubric Anchors use rubrics to guide LLM-based reward models, whereas we compile rubrics into executable code, removing the LLM inference layer entirely.

- **vs. VerIF/AutoIF**: These methods explicitly focus on verifiable constraints, while we extend the programmatic reward paradigm to genuinely subjective open-domain tasks such as creative writing and dialogue.

We position our work as providing a practical efficiency-performance trade-off for scenarios where computational resources are limited or rapid iteration is required, rather than claiming to replace all existing alignment methods. We believe this represents a complementary direction in the landscape of RL-based alignment techniques and hope it opens new avenues for future research.

We hope this clarification addresses the reviewers' concerns and welcome further discussion.

---

### Note · Authors · 2026-01-05

I have read and agree with the venue's withdrawal policy on behalf of myself and my co-authors.